# Order among chaos: High throughput MYCroplanters can distinguish interacting drivers of host infection in a highly stochastic system

**Melissa Y. Chen**[1]☽*, **Leah M. Fulton**[1]☽, **Ivie Huang**[1], **Aileen Liman**[1], **Sarzana S. Hossain**[1], **Corri D. Hamilton**[1], **Siyu Song**[1], **Quentin Geissmann**[1,2], **Kayla C. King**[1,3,4], **Cara H. Haney**[1,5]*

1 Department of Microbiology and Immunology, The University of British Columbia, Vancouver, Canada, 2 Center for Quantitative Genetics and Genomics, Aarhus University, Aarhus, Denmark, 3 Department of Zoology, The University of British Columbia, Vancouver, Canada, 4 Department of Biology, University of Oxford, Oxford, United Kingdom, 5 Department of Biological Sciences, University of Pittsburgh, Pittsburgh, Pennsylvania, United States of America

☽ These authors contributed equally to this work.
* melissa.c1010@gmail.com (MYC); chh333@pitt.edu (CHH)

**Data Availability Statement:** The datasets (raw images and 3D printing files) supporting the

## Abstract

The likelihood that a host will be susceptible to infection is influenced by the interaction of diverse biotic and abiotic factors. As a result, substantial experimental replication and scalability are required to identify the contributions of and interactions between the host, the environment, and biotic factors such as the microbiome. For example, pathogen infection success is known to vary by host genotype, bacterial strain identity and dose, and pathogen dose. Elucidating the interactions between these factors *in vivo* has been challenging because testing combinations of these variables quickly becomes experimentally intractable. Here, we describe a novel high throughput plant growth system (MYCroplanters) to test how multiple host, non-pathogenic bacteria, and pathogen variables predict host health. Using an Arabidopsis-*Pseudomonas* host-microbe model, we found that host genotype and bacterial strain order of arrival predict host susceptibility to infection, but pathogen and non-pathogenic bacterial dose can overwhelm these effects. Host susceptibility to infection is therefore driven by complex interactions between multiple factors that can both mask and compensate for each other. However, regardless of host or inoculation conditions, the ratio of pathogen to non-pathogen emerged as a consistent correlate of disease. Our results demonstrate that high-throughput tools like MYCroplanters can isolate interacting drivers of host susceptibility to disease. Increasing the scale at which we can screen drivers of disease, such as microbiome community structure, will facilitate both disease predictions and treatments for medicine and agricultural applications.

conclusions of this article are available in the Dryad repository (https://doi.org/10.5061/dryad.w9ghx3fxd). Processed data, code for analysis and figure generation, and a copy of 3D printing files can be found on our GitHub (https://github.com/mech3132/mycroplanter).

**Funding:** This work was supported by the following funding sources. The NSERC Banting Postdoctoral Fellowship (202209BPF-489437-BNE-CAAA-91388) (https://www.nserc-crsng.gc.ca/Students-Etudiants/PD-NP/Banting-Banting_eng.asp)) provided stipend support to MYC. The Human Frontier Science Program Organization (LT000325/2019) (https://www.hfsp.org/) provided salary support to QG. The NSERC Canada Graduate Scholarships – Master's (CGS M) award (https://www.nserc-crsng.gc.ca/students-etudiants/pg-cs/cgsm-bescm_eng.asp) provided stipend support to LMF. The Work Learn International Undergraduate Research Awards (WLIURA) (https://students.ubc.ca/career/campus-experiences/undergraduate-research/worklearn-international-undergraduate-research-awards) provided salary support to AL. KCK was funded by an NSERC Canada Excellence Research Chair (https://www.cerc.gc.ca/home-accueil-eng.aspx). This work was supported by an NSERC Discovery Grant and Accelerator Award (no. NSERC-RGPIN-2021-03587) (https://www.nserc-crsng.gc.ca/professors-professeurs/grants-subs/dgigppsigp_eng.asp) awarded to CHH. The Canada Research Chair salary award (https://www.chairs-chaires.gc.ca/home-accueil-eng.aspx) provided salary support to CHH. Funders did not play any role in study design, data collection and analysis, decision to publish, or preparation of the manuscript.

**Competing interests:** The authors have declared that no competing interests exist.

## Author summary

Whether or not hosts are susceptible to pathogen infection depends on complex interactions between the host and the environment. To rapidly and scalably test the contributions of host, non-pathogen, pathogen, and environment in predicting plant disease, we created the MYCroplanter system. MYCroplanters are a high-throughput plant growth platform that allows cultivation of the model plant *Arabidopsis thaliana* in 96-well arrays. Our tool is compatible with commercially available equipment and consumables, which means it can be used in tandem with high-throughput molecular tools like liquid handling robots and plate readers to design complex experimental set-ups. We use this system to demonstrate how pathogen establishment success is dependent on nuanced interactions between multiple host, non-pathogen, and pathogen variables. The sensitivity and scalability of our system offer the field of host-pathogen interactions an affordable way to conduct large-scale screens. The accessibility and flexibility of this tool will facilitate a new wave of research in predicting host health and disease, which can be leveraged to inform host treatment strategies.

## Introduction

Host susceptibility to disease is the result of complex interactions between host, pathogen, and the environment. Recent evidence has highlighted the role of the microbiome in protecting hosts from disease, adding another layer of complexity to classic disease models. The role of microbiota in protecting hosts from disease depends on complex interactions between host genotype [1,2], microbiota strain identity [3–6], and pathogen dose [7]. Host, pathogen, and microbiota traits also interact to alter the likelihood of infection [8–10]; consequentially, the effects of one variable are highly dependent on other components in the system. Quantifying the impacts of interactions between multiple variables on host susceptibility to disease is essential to predicting disease.

Inferring cause-effect relationships from host interactions with biotic and abiotic factors requires contending with a high degree of dimensionality and stochasticity in complex systems [11–13]. For instance, host susceptibility to pathogen infection is affected by host-environment interactions [14]. Furthermore, host-associated microbiomes contain up to tens of thousands of members [12] that interact with each other, the host, and the environment. Subtle differences in microbial community composition may interact with differences in host genotype [15] or abiotic conditions [16,17] to affect infection risk. Factorially testing multiple potential interacting variables that affect host health quickly becomes experimentally intractable.

Plants offer an attractive system to pioneer high-throughput platforms to dissect interacting predictors of infection. In particular, *Arabidopsis thaliana* is a small species of plant that has an extensive history as a model system [18] for both disease and microbiome research. Additionally, a previously established *Pseudomonas*-Arabidopsis system presents a unique opportunity to isolate interacting genetic and ecological mechanisms that may drive the likelihood of disease [19,20]. In this system, some but not all non-pathogenic *Pseudomonas fluorescens* strains protect plants against a closely related, opportunistic pathogen, *Pseudomonas fluorescens* N2C3 [19,20]. Varying bacterial strains and initial inoculation doses have previously been shown to affect the likelihood of plant disease [19]. This simplified and established model is therefore ideal for investigating the effects of multiple variables on pathogen establishment and plant disease.

To test multiple interacting variables that may affect plant disease susceptibility, we designed a novel high-throughput plant cultivation system, MYCroplanters. MYCroplanters are 3D-printed miniature planters that allow for cost-effective gnotobiotic cultivation of Arabidopsis seedlings in 96-well arrays and quantification of both plant health and bacterial abundance. MYCroplanters allow rapid screening of treatments because they are compatible with liquid-handling robots and multichannel pipettes. Plants are scanned with a commercial scanner to extract colour and size information, which are combined as metrics of health or disease (represented by the probability of plants remaining healthy). Additionally, compatibility with 96-well plate readers allows us to quantify pathogen or non-pathogenic bacterial abundance using fluorescently labelled isolates.

We use MYCroplanters to isolate the effects of multiple interacting host, pathogen, and non-pathogenic bacterial variables on the likelihood of plants remaining healthy. First, we test the interaction between (1) pathogen and non-pathogenic bacterial inoculation dose across bacterial strains, and (2) pathogen doses across plant genotypes and bacterial strains. We found that plant genotype and bacterial strain identity were strong predictors of plant health, and that altering ratios of pathogen to non-pathogen could mask effects of host genotype. This finding suggests the relative abundance of pathogenic to non-pathogenic bacteria may be an important determinant for plant health. We therefore quantified the relationship between pathogen and non-pathogenic bacterial abundance, and plant health by tracking fluorescently labeled pathogen and non-pathogenic bacterial strains. We inoculated plants with different pathogenic to non-pathogenic bacterial ratios and altered their arrival order. We found that the ratio of pathogen to non-pathogen was a stronger predictor of plant health than plant genotype, non-pathogenic bacterial strain identity or dose, strain arrival order, or pathogen dose. These data indicate that with sufficient throughput, multiple interacting variables can be isolated in highly complex systems. Dissecting contributions of individual variables can determine which are most important in predicting the likelihood of host health or disease, which will facilitate disease prevention for agriculture [21] or medicine [22].

## Results

### MYCroplanters provide reliable plant-bacterial interaction screening at high-throughput capacity

We first conducted proof-of-viability experiments for MYCroplanters (Fig 1) by replicating previous findings from a *Pseudomonas*-Arabidopsis model system [19,20]. In this model, some non-pathogenic *Pseudomonas* strains can protect plants from an opportunistic pathogen *Pseudomonas fluorescens* N2C3. We treated plants with pathogenic N2C3, beneficial WCS365, and both strains together (Fig 1b and S1 Fig). We confirmed phenotypically, that N2C3 caused disease, WCS365 did not, and both strains together resulted in mostly healthy plants (Fig 1b). To quantify the probability of plant health, we developed a custom Python script to extract pixel data from scanned plant images and created a "Health score" to rate plants on their degree of disease based on plant colour and size (S2A and S2B Fig Health Score data derived from experiment outlined in S3 Fig). We found that the health score could detect quantifiable differences in the phenotypes of Mock-treated and WCS365-treated plants from N2C3-treated plants (Fig 1C). As our health scores do not have any direct biological interpretation, we decided to describe plant health or disease as a bimodal response variable (healthy/diseased) to eliminate over-interpretation of plant colour and size as a proxy for health (Fig 1D and 1E, and S2E Fig). Thus, binary outcomes are reported for subsequent experiments (Fig 1E and onwards).

We then inoculated plants with five *Pseudomonas* spp. strains that have diverse functional relationships with Arabidopsis (WCS365, CHA0, CH267, Pf5, and N2C3) at different

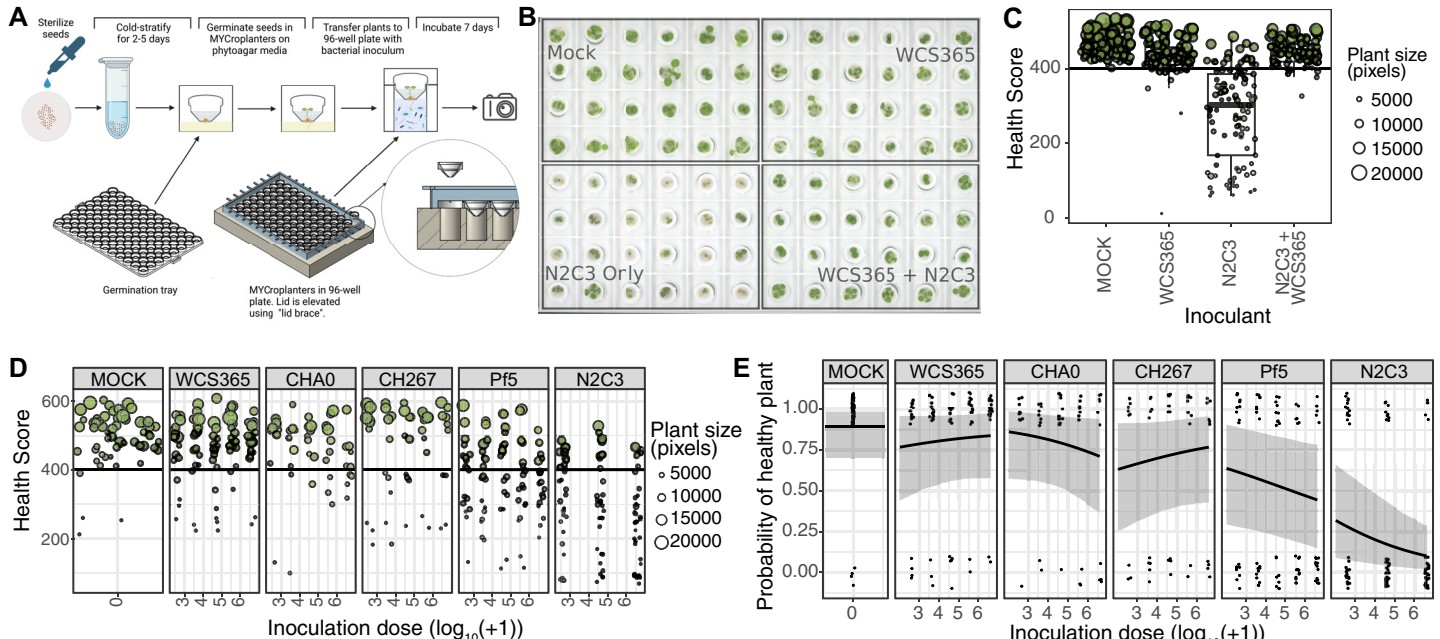

**Fig 1. MYCroplanters can quantify the interaction between pathogenic and non-pathogenic bacteria and their effects on plant health.** (a) Schematic showing growth set-up for Arabidopsis seedlings in MYCroplanters. Sterilized seeds are germinated inside MYCroplanters sitting on 0.5% phytoagar with 1/2MS 1/2MES 2% sucrose, and then transferred to a 96-well hydroponic system containing pre-diluted bacterial cultures. Plants are scanned after 1 week of incubation with bacteria and images are processed using a custom Python script to extract pixel data. Plants are assigned "health scores" based on their colour and size (Hue*saturation*$\log_{10}$(Plant pixels + 1)*100). Figure panel created in BioRender (b) A sample image from a plate containing Mock, *Pseudomonas* spp. N2C3 (pathogen), WCS365 (non-pathogenic), and N2C3 + WCS365 treatments. (c) Plant health scores extracted from image (b) and others, showing WCS365 protects against N2C3. Each dot is a single plant. Boxplots show the average (middle line), interquartile range (box), and last quartiles (whiskers) of $\log_2$ fold change values. (d) Plant health scores from 5 different *Pseudomonas* strains (and a mock treatment) on Arabidopsis health across 3–5 inoculation doses (3 for N2C3; 5 for all other strains). Units for inoculation dose are number of cells ($\log_{10}$) as estimated by absorbance at 600nm). Data shows that N2C3 is pathogenic, whereas strains WCS365, CHA0, CH267, and Pf5 are largely non-pathogenic. Each dot represents the health score of a single plant. For (c-d), dots are coloured by the average colour from pixel images (averaged RGB scores) and sized relative to overall plant size (number of pixels). The horizontal black line represents the health score cut-off distinguishing healthy from diseased plants (>400), which was derived from an average cutoff value from human-ranked plant images (S2 Fig). (e) Figure showing data from (d) converted into binary health/disease scores. Each dot represents a single plant. Black lines with ribbons are Bayesian model predictions with 95% prediction intervals, respectively. Plant age was included as a fixed effect, and experimental plate was included as a random effect in the model.

inoculation doses (Fig 1D and S3 Fig). By measuring plant health scores, we confirmed that consistent with previous reports [19], N2C3 was pathogenic and that other strains were largely non-pathogenic (Fig 1D and 1E). Our system also revealed several subtle qualities about each strain that were previously challenging to quantify. Using a Bayesian modelling approach, we estimated the effect sizes and 95% credible intervals (reported as: median [lower 2.5%, upper 97.5%]) of different inoculation treatments and concentrations on the probability of plants remaining healthy. As Bayesian approaches do not use *p* value cut-offs, we interpret effects as "strong" when 95% credible intervals are non-overlapping with zero [23]. We found that the presence of N2C3 (regardless of inoculation dose) reduced the probability of plant health significantly (N2C3 presence = effect size -2.57, 95% credible interval [-3.81, -1.40]), while the concentration of N2C3 inoculum had only a small effect (N2C3 cell load = -0.34 [-0.54, -0.14], Fig 1E and S4 Fig). N2C3 also reached similar bacterial loads based on absorbance readings after 1 week regardless of inoculation concentration (S5 Fig), suggesting that N2C3 infection is a relatively binary outcome in monoculture. Additionally, we found that regardless of dose, WCS365 (WCS365 presence = -1.24 [-2.94, 0.44]) and CHA0 (CHA0 presence = 0.21 [-1.91, 2.47], Fig 1E and S4 Fig) presence in monoculture have no significant effect on plant health. In

contrast, the presence (but not necessarily concentration) of CH267 (CH267 presence = -2.09 [-3.92, -0.26]) and Pf5 (Pf5 presence = -1.36 [-2.87, 0.09]) slightly decreased the probability of plants remaining healthy (Fig 1D and S4 Fig). This finding suggests that some apparently non-pathogenic bacteria may inflict a subtle disease burden on plant hosts when colonizing alone and that MYCroplanters can rapidly quantify these effects.

## Plant disease is best predicted by non-pathogen to pathogen ratio, but modulated by inoculation dose

Next, we tested whether MYCroplanters could provide sufficient resolution to distinguish the effects of multiple interacting variables on the likelihood of plants developing disease. A previous study found that bacteria became more effective at protecting plants against the N2C3 pathogen when the N2C3 dose decreased [19]. We asked whether loss of protection was due to differences in the absolute abundance of pathogen (regardless of non-pathogenic bacterial dose), or if the ratio between non-pathogen and pathogen was responsible for the differences in plant health. To test the effect of non-pathogen to pathogen ratio across bacterial strains, we conducted a screen of 24 combinations of different non-pathogen (0, $10^3$, $10^4$, $10^5$, $10^6$, $10^7$ cells) and pathogen (0, $10^3$, $10^5$, $10^7$ cells) cell doses (S6 Fig). We tested four non-pathogenic bacterial strains (WCS365, CHA0, CH267, Pf5) to determine whether inoculation ratio-dependent plant disease was a consistent feature across bacterial strains.

We found that bacterial strain, non-pathogen to pathogen inoculation ratios and absolute inoculation doses all contributed to the probability of plants remaining healthy (Fig 2 and S7 Fig). However, strain identity and non-pathogen to pathogen inoculation ratio were better predictors of plant health than pathogen inoculation dose (Fig 2). Higher pathogen doses required higher non-pathogen doses to maintain protective phenotypes for WCS365 (WCS365:N2C3 cell loads = 0.044 [0.02, 0.069]), CHA0 (CHA0:N2C3 cell loads = 0.068 [0.04, 0.10]), and Pf5 (Pf5:N2C3 cell loads = 0.085 [0.062, 0.11], Fig 2 and S7 Fig). This indicates that

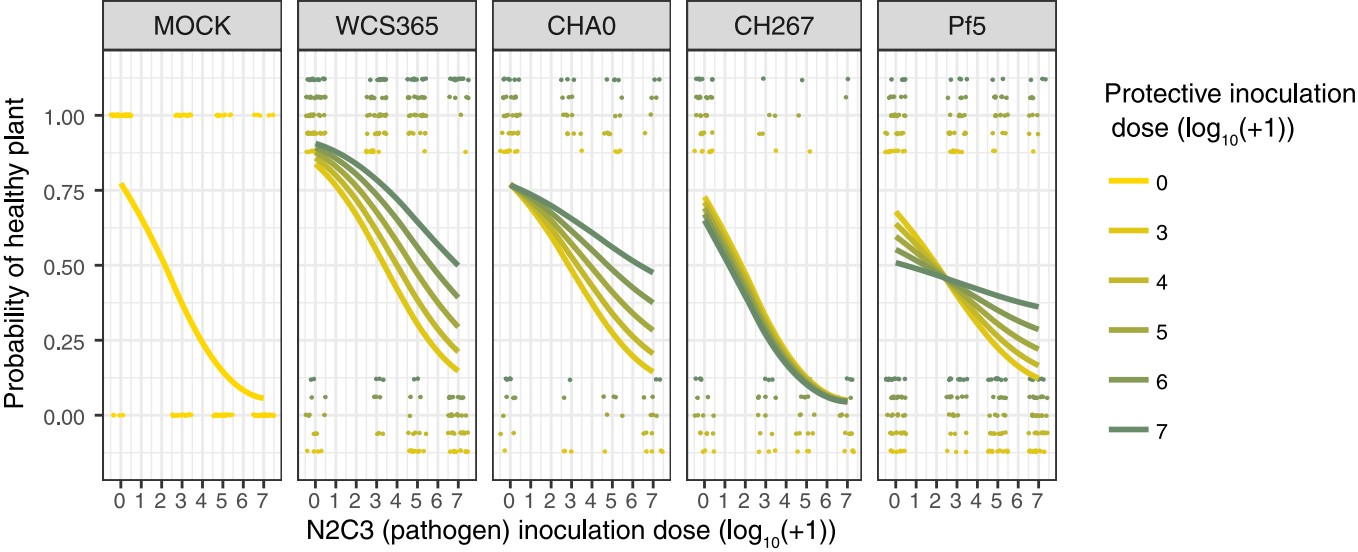

**Fig 2. Plant health depends on non-pathogen to pathogen ratio.** The probability of a plant remaining healthy (y-axis) decreases with pathogen dose (x-axis). Increasing non-pathogenic bacterial strain dose (yellow to green lines) improves the probability of a healthy plant in strains *Pseudomonas* spp. WCS365, CHA0, and Pf5. Each point represents a single plant, and data are offset vertically by protective inoculation dose to better visualize the effect of different bacterial doses. Units for cell dose are number of cells ($log_{10}$) as estimated by absorbance at 600nm. Points are jittered horizontally. Lines represent mean Bayesian predictions of the probability of plants remaining healthy under each treatment.

the ratio of non-pathogenic to pathogenic bacteria rather than dose of either is a stronger predictor of plant health. Strain CH267, which was previously identified as non-protective [19], did not protect against N2C3 at any ratio (CH267:N2C3 cell loads = 0.0062 [-0.027, 0.037], Fig 2 and S7 Fig). Further, leave-one-out (LOO) comparisons of models with and without the inoculation ratio or dose terms showed that the best model included both ratios and doses as a predictor, while the second best model consisted of only inoculation ratio (ELPD difference = -19.3, SE difference = 6.5), rather than only inoculation dose (ELPD difference = -29.4, SE difference = -7.8). Thus, for protective bacteria, the ratio of non-pathogenic to pathogenic bacteria is the strongest determinant of plant health.

Inoculation doses of bacteria had different effects on plant health for different bacterial strains (Fig 2 and S7 Fig). For example, we found that while increasing WCS365 inoculation dose improves the probability of plants remaining healthy (WCS365 cell load = 0.16 [0.058, 0.28], Fig 2 and S7 Fig), increasing Pf5 inoculation dose decreases plant health (Pf5 cell load = -0.21 [-0.30, -0.12], Fig 2 and S7 Fig) likely because of it acts as a weak pathogen alone. Inoculation doses therefore interact with bacterial strain to predict the likelihood of disease. These results suggest distinct mechanisms in the mode of protection between *Pseudomonas* strains Pf5 and WCS365.

## Plant host factors contribute to protection by non-pathogens

Plant developmental age and genotype affect microbiome-mediated resilience to biotic and abiotic stress [8,24,25]. We first asked whether MYCroplanters could quantify the interaction of plant age with pathogenic and non-pathogenic bacterial strain on plant health (S8 Fig). We found that plants had a lower probability of developing disease at older ages (effect of plant age = 0.94 [0.75, 1.14]; S8 Fig). Additionally, there was an interaction between pathogen presence, plant age and bacterial strain on plant health (WCS365 = 0.36 [0.19, 0.53], CHA0 = 0.39 [0.21, 0.57], and Pf5 (0.5 [0.34, 0.67]); S8A Fig), indicating that older plants were more easily protected against N2C3 than younger plants, even after accounting for the effect of age or N2C3 alone. As we found that non-pathogenic bacteria are more effective at improving plant health on older plants, plants may mediate the interaction between beneficial bacteria and pathogens in favour of the beneficial strain.

Plant immune function can distinguish between closely related beneficial and pathogenic strains [26], and so differences in plant immune function between 5-day-old and 7-day-old seedlings may result in different outcomes of host-microbe-pathogen interactions. We hypothesised that older plants may have a more developed immune system [27], which may allow plants to better distinguish between or react to pathogenic and non-pathogenic bacteria. The improved ability to detect and respond to strains of different lifestyles may therefore allow plants to alter competition between strains in favour of the non-pathogenic strains. To test the interaction of bacterial strain with host immunity, we tested the protective ability of *Pseudomonas* spp. WCS365, CHA0, and Pf5 on *bik1* and *bak1*, *bkk1*, *cerk1 (bbc)* plant immune mutants (see S9 Fig for experimental design). We found that there was no interaction of plant genotype with pathogen dose on the likelihood of plant disease (*bbc* = 0.52 [-1.10, 2.17], *bik1* = -0.61 [-2.42, 1.17], Fig 3). This indicates that the immune mutants were equally susceptible to N2C3 relative to wildtype plants irrespective of pathogen dose. To determine whether predicted outcomes between different treatments were statistically distinguishable from each other, we compared prediction intervals (simulated health ranges) between plant genotype treatments. We found that when inoculated in 1:1 ratios of non-pathogen to pathogen, the protective ability of WCS365 and CHA0 decreased on *bbc* plant mutants compared to the WT Col-0 genotype (non-overlapping 95% prediction intervals for WCS365 Col-0 = [0.74,0.99] vs. WCS365 *bbc* =

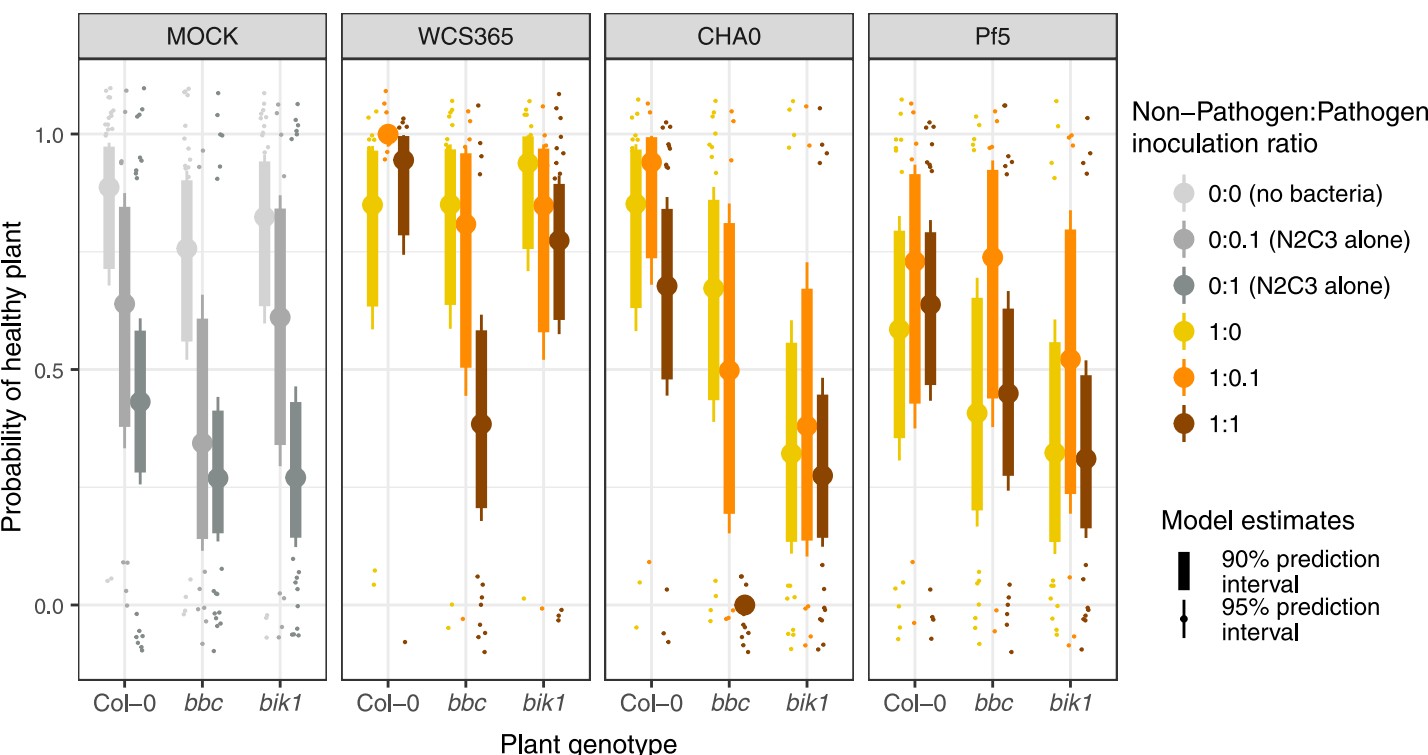

**Fig 3. Plant health depends on the interaction between plant immunity, and non-pathogen to pathogen ratio.** The probability of plants remaining healthy (y-axis) across different plant mutants (x-axis) and inoculation ratios (colours). Col-0 is the WT plant, and *bbc* and *bik1* are pattern-triggered immunity immune (PTI) mutants. Plant health was decreased more for *bbc* mutants than Col-0 plants in N2C3:WCS365 and N2C3:CHA0 treatments when inoculation ratios are 1:1. Pf5 reduced the plant health alone in *bbc* mutants, so there was no detectable difference between Col-0 and *bbc* plants when co-inoculated with N2C3. The loss of protection in *bbc* mutants for WCS365 and CHA0 can be partially rescued by decreasing the non-pathogen to pathogen ratio from 1:1 to 1:0.1.

[0.18, 0.62]; CHA0 Col-0 = [0.44, 0.87] vs. CHA0 *bbc* = [7.7e-6, 0.02]) (Fig 3). WCS365 protected against N2C3 on *bik1* (overlapping 95% prediction intervals, Col-0 = [0.74, 0.997 vs. *bik1* = [0.58, 0.91]) but CHA0 did not protect as well on *bik1* (marginally overlapping 95% prediction intervals, Col-0 = [0.44,0.87] vs. *bik1* = [0.12,0.48]) (Fig 3). Pf5 alone decreased the probability of plants remaining healthy in both *bbc* and *bik1* mutants, suggesting these plant mutants may be more susceptible to the pathogenic effects of Pf5. There were no detectable differences in plant health with or without N2C3 on Pf5-treated plants (Fig 3). These results suggest that intervention by the plant immune system favours non-pathogenic strains and that plant immune function interacts with bacterial strain to determine the outcome of host-microbe interactions.

Interventions by the plant through immune function are not necessarily required for protection as we found that decreasing pathogen dose restored protection by some bacterial strains. Higher WCS365 and CHA0 to pathogen ratios (1:0.1, non-pathogen:N2C3) improved protection against N2C3 on *bbc* mutants such that plant health was indistinguishable from no pathogen controls (overlapping 95% prediction intervals for WCS365 10:1 = [0.44, 0.97] vs. WCS365 alone = [0.58, 0.97]; CHA0 10:1 = [0.15,0.85] vs. CHA0 alone = [0.39, 0.89]; Fig 3). The result that increasing non-pathogen to pathogen ratio restores protection indicates that there are likely additional ecological mechanisms driving the outcome of plant-bacterial interactions that are independent of host plant immune function.

## Inoculation ratios predict community dominance

Bacterial inoculation ratio (Fig 2) and host genotype (Fig 3 and S8 Fig) both influenced plant health, but we found a substantial amount of variation in the probability of plants remaining healthy even within these treatments. As variation in plant health may be a result of variable community structure [28] or variable interactions between community structure and other host factors [1], we decided to focus on variability associated with non-pathogen and pathogen ratios. We asked whether varying initial non-pathogen to pathogen ratios resulted in a predictable final structure of a 2-member community.

We hypothesised that inoculation ratios predicted final rhizosphere community structure variation, and that there would be variability in final non-pathogen to pathogen ratio when initial inoculation ratios were near 1:1. We conducted a WCS365-N2C3 competition experiment with WCS365 and N2C3 expressing mNeonGreen (Neon) or E2-Crimson (Crimson) (reciprocally) from the same rhizosphere stable plasmid backbone under a constitutive promoter [29]. N2C3 and WCS365 were inoculated at different concentrations (see S10 Fig for experimental design) and their $\log_2$-fold ratios at the end of the experiment (after 1 week of growth on the plant) were quantified using a fluorescence plate reader.

In asymmetric inoculation treatments (where WCS365:N2C3 ratios were not 1:1), dominance of the final 2-member rhizosphere community was strongly predicted by which strain had the larger starting population (Fig 4). When N2C3 was inoculated in greater abundance than WCS365, N2C3 was more abundant within the final community ($\log_2$FC [min, max] = -6.43 [-12.9, 0.827], Fig 4). In contrast, when WCS365 was inoculated in greater abundance, WCS365 was more abundant than N2C3 ($\log_2$FC = 6.85 [3.53,14.3], Fig 4). Thus, the dominant strain in uneven relative starting populations strongly predicts which strain will dominate in the final 2-member community.

We predicted that treatments with approximately equal starting populations would stochastically fix in either direction in approximately equal proportions. Indeed, we found that treatments with 1:1 inoculation ratios varied greatly in which strain dominated the final 2-member community (Fig 4). Final community composition varied from strong N2C3 dominance ($\log_2$FC = -5.29) to strong WCS365 dominance ($\log_2$FC = 9.58). Thus, we find that community assembly is seemingly stochastic when inoculation ratios are near 1:1. Inoculating two strains in similar proportions therefore leads to highly variable assembly trajectories, and results in highly variable assembly of a two-member community.

## Order of arrival during early colonisation drives variation in strain dominance

Our fluorescence experiment revealed substantial variation in community dominance when strains were inoculated in 1:1 ratios. We next wondered what deterministic or stochastic processes were responsible for variation in competition outcomes. We hypothesised that variation in community dominance could be a result of either true stochastic processes (neutral assembly, fixation through drift), or stochasticity in who arrives first (priority effects) at the microscopic level, leading to deterministic outcomes after initial colonization. To differentiate these two possibilities, we conducted an experiment where N2C3 and WCS365 (with either Crimson or Neon fluorescent markers) were inoculated in different orders. Introduction of the second strain was delayed by 3 seconds (dipped in first for 3 seconds, then immediately transferred to a second well), 3 hours, 6 hours, 24 hours, or 48 hours (S11 Fig). Staggered inoculations were accomplished by placing plants in a well with the first strain and then transferring plants to a second plate with wells containing only the second strain ("well-to-well" method). We also included an additional treatment where plants were dipped in one strain and then dipped in

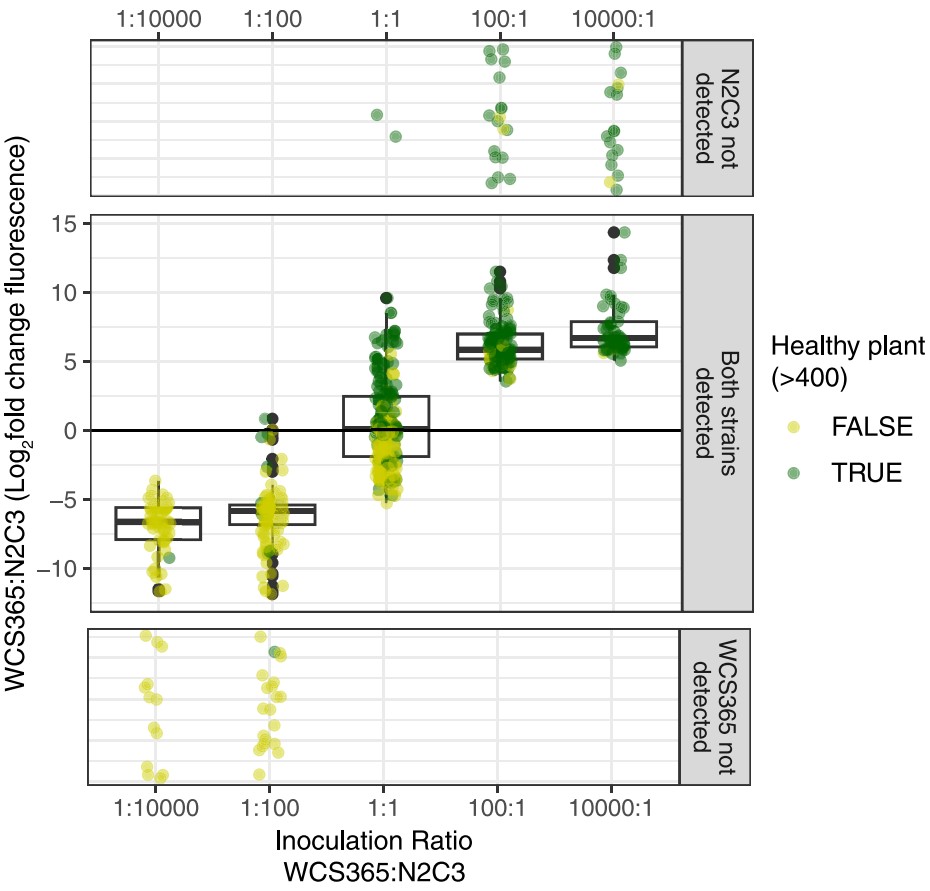

**Fig 4. Bacterial inoculation ratio predicts which strain dominates the 2-member community.** Plants were inoculated with non-pathogenic WCS365 and pathogenic N2C3, each labelled (reciprocally) with either Neon or Crimson plasmids. Inoculation ratios were varied, and final relative fluorescence was measured for the corresponding plant. Inoculation ratios (x-axis) strongly predict whether WCS365 or N2C3 dominate the final 2-member community (y-axis; $\log_2$ fold change of fluorescence values for WCS365 and N2C3). In some treatments, one of the strains was below the threshold of detection so we include these points in the top and bottom panels separately (points are jittered randomly) because ratio calculations would be indeterminate (cannot log or divide by zero). Boxplots show the average (middle line), interquartile range (box), and last quartiles (whiskers) of $\log_2$ fold change values, where positive numbers means more WCS365. Points are coloured by plant health (healthy / not healthy).

the second strain before being placed in a well containing sterile plant media ("double dip" method). This was done to ensure that the second strain did not have a substantial cell dose advantage in the 3 seconds time delay treatment.

Across well-to-well priority effect treatments, invasion success by the second strain decreased with time between introductions (Fig 5A). Plants treated with WCS365 first for at least 6 hours were consistently dominated by WCS365. N2C3 had a similar but weaker priority effect. Surprisingly, with the double-dip method we found that strains inoculated 3 seconds before a second strain had a competitive advantage in the rhizosphere (Fig 5B). Plants that were dipped in WCS365 and then in N2C3 were more strongly dominated by WCS365 than plants dipped in the reverse order (WCS365 dominance in WCS365-first treatments = 3.12 [2.24,3.98]; WCS365 dominance in N2C3-first treatments = -1.76 [-2.41, -1.12], Fig 5B). We confirmed that cell dose on 3 second dipped plants were sufficient for establishment on the rhizosphere by including controls where plants were dipped in monocultures then dipped in sterile media (Fig 5B). Thus, we find that even subtle spatiotemporal advantages during early

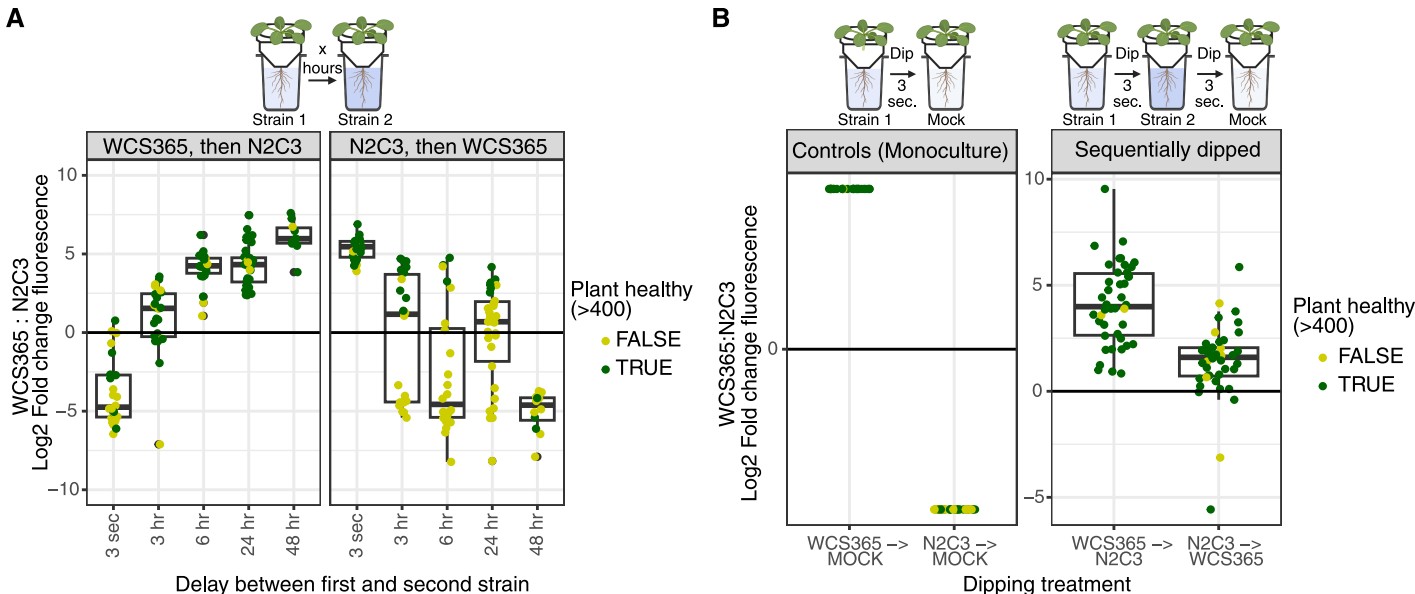

**Fig 5. Priority effects during early root colonization shape bacterial competition outcomes.** Plants were inoculated with WCS365 and pathogenic N2C3, each labelled (reciprocally) with either Neon or Crimson plasmids, and inoculated with different time delays. For the "well-to-well" treatments (a) plants were dipped in the first strain and placed in a well filled with the second strain suspended in plant media. For the "double dip" treatment (b), plants were dipped in the first strain for 3 seconds; dipped in the second strain for 3 seconds; and placed in a well with sterile media. The relative fluorescence of each strain was measured using a plate reader, and plant health was quantified for each corresponding plant. (a) The first strain has a colonization advantage over the second strain. The strength of priority effects increases with time. (b) WCS365 dominates the community more strongly when plants are dipped in WCS365 before N2C3, than when plants are dipped in N2C3 before WCS365. For (a) and (b), boxplots show the average (middle line), interquartile range (box) and last quartiles (whiskers) of log$_2$ fold change values, where positive numbers mean more WCS365. Points are coloured by plant health (healthy/not healthy). Plant diagrams created in BioRender https://BioRender.com/g72k241.

colonization have profound effects on the trajectory of community assembly in the rhizosphere.

We observed that the first bacterial strain we inoculated plants with increased in cell number on the root significantly between 0 and 24 hours (S12 Fig) based on CFU counts on roots. Because we found strong effects of non-pathogen to pathogen ratio on community assembly (Fig 4), the increase in cell number of strain 1 could be the result of an increase in ratio rather than arrival order. Thus, we proposed two alternative hypotheses as to why greater delays in the second strain resulted in less invasion success. First, we hypothesised that invasion success was proportional to the ratio of the first strain (on the root) and the second strain (in the well) at time of second strain inoculation. This would support a "neutral assembly" mechanism, where final community composition is proportional to population ratios (Fig 6, solid black lines; slope estimated from simultaneous inoculation controls). Alternatively, we hypothesised that the establishment of the first strain on the root might physically preclude the establishment of the second strain as it saturates niche space. This would support a "priority effects" mechanism, where strains given a temporal advantage will always dominate the community because of its colonization advantage (Fig 6, dotted black lines). We conducted an additional priority effects experiment where cell doses of the second strain were varied, to determine whether invasion success favoured the first strain after controlling for differences in cell numbers.

We found that when the dose of the first and second strains are similar, early colonization imparts strong benefits on the first strain (WCS365-first advantage = 3.31 [2.74,3.89], N2C3-first advantage = -1.70 [-2.10,-1.29], Fig 6). However, increasing the inoculation dose of

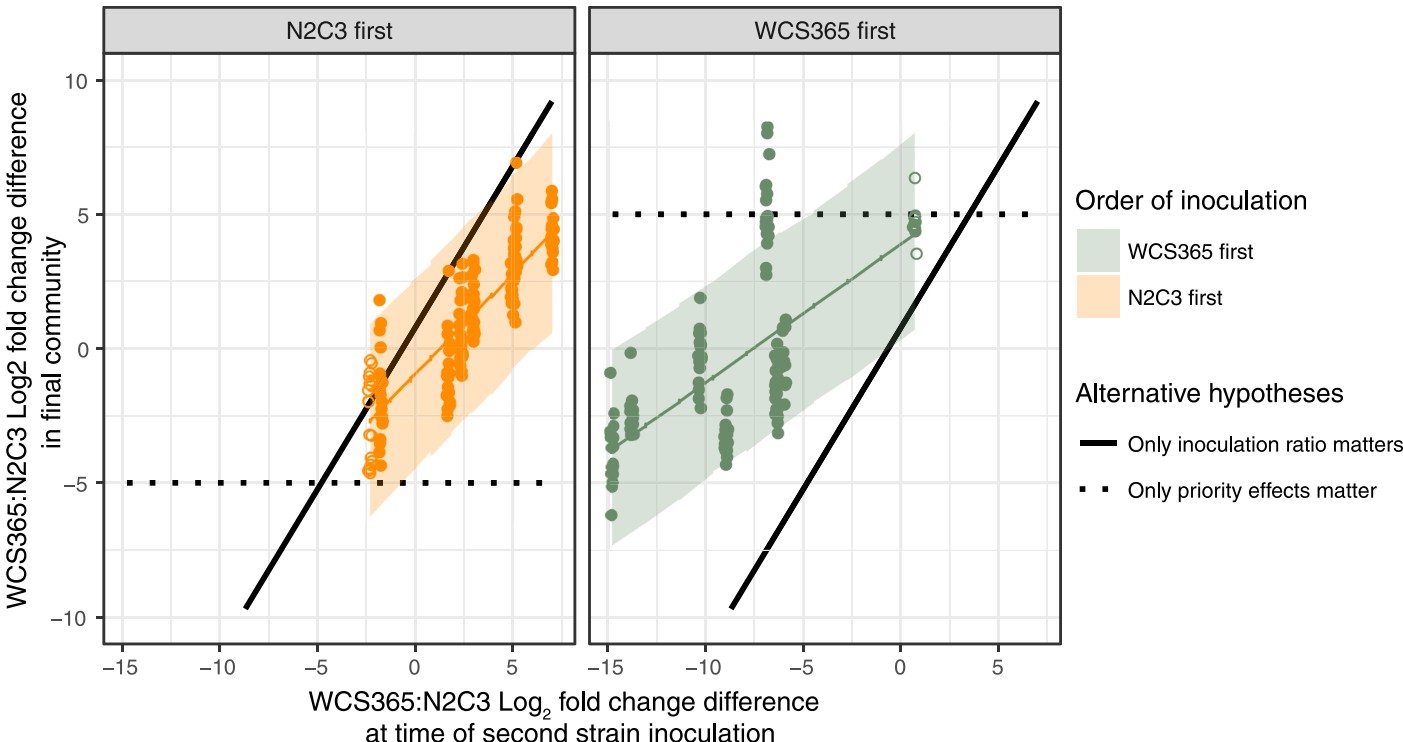

**Fig 6. Both priority effects and inoculation load influence bacterial assembly trajectories.** Plants were inoculated with the first strain and incubated for either 6 or 24h then moved to the second strain at different doses. If priority effects were the dominant force shaping microbial communities, then the first strain was expected to dominate regardless of relative abundance of either strain (dotted black lines). If population ratios were the dominant force shaping the 2-member bacterial community, then the final community composition was expected to correlate with starting population ratios (solid black lines). Fold-change differences for inoculation ratios were derived from CFU counts on the root and in the well. Fold-change differences for the final community composition were derived from fluorescence measurements. Each dot indicates one plant; lines are predictions from a Bayesian regression and ribbons are 95% prediction intervals. The solid black line is the median Bayesian posterior estimate of the relationship between inoculation ratio and final community ratio based on concurrent "simultaneous inoculation" treatments, where strains were introduced at the same time (see S11 Fig for experimental set-up). Total community dominance (dotted lines) was set to -5 (N2C3 dominant) and 5 (WCS365 dominant) as approximate estimates for a typical "fixed" community (see Figs 4 and 5). Closed dots indicate treatments where strain 2 was introduced at the 6h time point, whereas open dots indicate treatments where strain 2 was introduced at the 12h time point.

the second strain can overwhelm priority effects (Inoculation ratio = 1.20 [1.05, 1.35], Fig 6). This result suggests that two processes are occurring simultaneously. First, priority effects influence community assembly in the rhizosphere, and this effect is strongest when resident and invading population sizes are equal. Second, the inoculation load of invaders may over-whelm priority effects in certain conditions.

## A host affects the competition between non-pathogen and pathogenic bacteria

While microbial assembly outcomes predict host health, we find that the host environment also has profound effects on assembly of a 2-member community. That is, bacterial competition outcomes differ between *in vivo* versus *in vitro* environments (S13 Fig). We found that *in vitro*, WCS365 and N2C3 are evenly matched in terms of growth rate, carrying capacity, and competitive ability (S13 and S14 Figs). *In planta*, however, WCS365 gains a fitness advantage over the pathogen N2C3 (Figs 4 and 5). WCS365 is more likely to dominate the community in 1:1 simultaneous inoculations (Fig 4) and WCS365 more easily prevents N2C3 establishment than vice-versa (Fig 5). However, if strains WCS365 and N2C3 are competed on solid plant

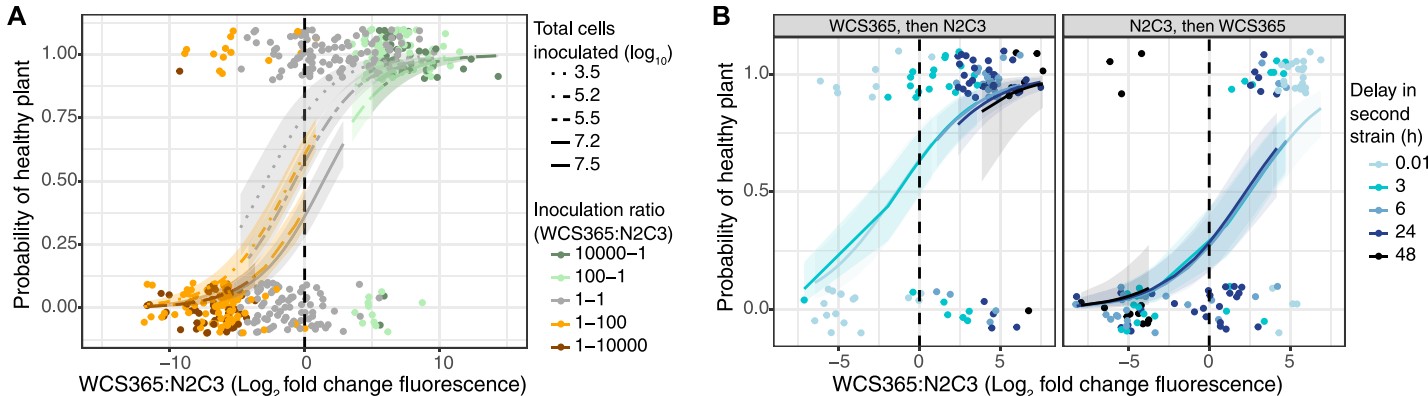

**Fig 7. Regardless of inoculation ratio, order of arrival, or delay in second strain introduction, plant health was strongly predicted by final community composition.** (a) The probability of plant health associated with Fig 4, where N2C3 and WCS365 were simultaneously inoculated at different doses and ratios. (b) Plant health associated with Fig 5B, where plants were inoculated with N2C3 and WCS365 in different orders with various time delays between the first and second strains. For (a-b), dots represent individual plants. Colours represent different variables in each experiment within treatments. Lines are median Bayesian predictions based on a Bernoulli model, while bands represent 95% prediction intervals.

media without the presence of plants, neither strain dominates the community, even when inoculated 24 hours apart (S13 Fig). These data show that, firstly, strain traits *in vitro* are not correlated with competitive ability in the rhizosphere. Secondly, screening competition outcomes *in vivo* is crucial to understanding drivers of bacterial community assembly in host contexts because the host environment alters bacterial interactions.

### Regardless of inoculation conditions, the likelihood of plants developing disease is predicted by non-pathogen to pathogen ratio in the rhizosphere

Above, we demonstrate that bacterial community assembly can be highly variable under certain conditions (e.g. simultaneous inoculation of equal population sizes), but that outcomes are a result of multiple interacting forces, including order of arrival, arrival delay, and inoculation dose. We then tested whether final composition of a 2-member community predicts the likelihood of plants remaining healthy. We hypothesised that regardless of the stochasticity involved in bacterial community assembly, the final community on the root, as measured by the ratio of WCS365 to N2C3, would predict plant health.

Indeed, we find that plant health is strongly correlated with final community structure regardless of inoculation conditions (Fig 7). Regardless of inoculation ratio (Fig 7A) or arrival order (Fig 7B), final community composition (derived by fluorescence ratios between both strains) is a strong predictor of plant health (Ratio experiment = 0.39 [0.27, 0.51]; Priority effects experiment = 0.41 [0.29, 0.55]). Further, model comparisons using leave-one-out (LOO) cross-validation (LOO.brmsfit) confirmed that final fluorescence ratio was a better predictor of plant health than inoculation ratio (ELPD difference = -27.3, SE difference = 7.4) or arrival order (ELPD difference = -24.3, SE difference = 8.2). The source of variability in plant health and disease is therefore a result of variability in community assembly.

## Discussion

Whether or not organisms will develop disease is a complex phenomenon dependent on multiple interacting host, microbiome, and pathogen variables [1,8,16,30]. We demonstrate that with enough power and replication, it is possible to resolve multiple interacting drivers of host health and disease in highly stochastic systems. Using a *Pseudomonas*-Arabidopsis model, we

find that variation in host disease is robustly linked to variation in final ratio of non-pathogenic to pathogenic bacteria in the final community. Further, community structure in a 2-member community is dependent on an array of interacting host by bacterial strain effects. Lastly, dynamics between non-pathogens and pathogens are host-context-dependent. These findings emphasize the importance of not only understanding the rules and drivers of bacterial community assembly, but doing so *in vivo*.

We find that competition outcomes between non-pathogens and pathogens within the rhizosphere are jointly predicted by multiple processes that can both negate and amplify each other's effects [31,32]. For example, plants with reduced immune functions are more susceptible to infection by the pathogen N2C3, but decreasing inoculation dose in the presence of non-pathogenic bacteria can partially rescue plants from infection (Fig 3). We also find that early-arriving strains have an advantage in the rhizosphere and are more likely to dominate the final bacterial community (Fig 5). However, increasing the inoculation dose of the second strain (analogous to increasing propagule pressure in invasion models [33]) can overwhelm priority effects (Fig 5). The interactive effects observed in our system highlight two important considerations when interpreting findings in microbiome research. First, many different drivers of microbiome assembly are likely operating simultaneously [31,32] and may therefore obscure clear cause-and-effect even in highly simplified systems. Second, the influence of individual drivers of bacterial community assembly may be masked by other components. Thus, contradictory findings between systems (e.g. priority effects were found to be important in one system, but not in another [34–36]) do not necessarily mean systems do not share rules for assembly: it may simply mean that the way rules manifest are context-dependent [37,38].

Phylogenetic and ecological backgrounds of non-pathogenic and pathogenic bacteria are also important in predicting host susceptibility to disease, and microbial community assembly [19,39]. Ecologically and genetically similar strains are more likely to reciprocally prevent establishment of the other [40]. Since we used a conspecific pathogenic and non-pathogenic strain pair, it is unsurprising to have found strong priority effects. Broadening the phylogenetic background of strains may alter the strength of priority effects during assembly [41,42]. Additionally, as communities are scaled up in complexity, order of arrival may impact some pairwise strain comparisons but not others. Order of arrival may therefore alter the trajectory of community succession without fully occluding new arrivals [42]. Although priority effects are a well-established and important deterministic force driving community assembly [35,36,42,43], effects can be overwhelmed by other drivers such as high community turnover or propagule pressure of an invader [41,44]. The lack of observed priority effects in certain conditions [37,41] does not detract from their legitimacy; the inconsistent detection of priority effects simply indicates a condition (e.g. environmental context, propagule pressure, host genotype) under which it influences final community composition. To fully resolve the contexts and situations in which priority effects are relevant to pathogen establishment across a broader range of bacterial strains to reflect the true complexity of communities, high throughput-platforms like MYCroplanters will be invaluable due to their scalable experimental capacity.

## Conclusions

Using the ultra-high replication of MYCroplanters, we demonstrate that host health is ultimately predicted by microbial assembly outcomes. Final microbial community structure can be influenced by a combination of deterministic and stochastic processes, including host genotype, order of arrival, and inoculation load. These factors can negate, overwhelm, or compensate for each other. Our findings therefore emphasize that while microbiome functions are ultimately predictable, the forces driving microbial assembly are layered and nuanced. Future

research should embrace the seemingly chaotic qualities of microbial communities and leverage the current wave of novel high-throughput tools to better resolve the underlying drivers of stochasticity in difficult datasets. Doing so will facilitate the translation of lab results into practical and effective approaches in microbiome engineering that can help combat pathogens in diverse host systems [21,22].

## Methods

### The MYCroplanter system

MYCroplanters are custom 3D printed polylactic acid (PLA) micro-planters that grow *Arabidopsis thaliana* seedlings in 96-well plates. Sterile MYCroplanters (see "Sterilization of equipment" section below) were placed in a 8x12 array germination tray on top of solid plant media containing 1/2X MS (Phytotech Labs PID:M519), 0.5% MES (VWR CAS#4432-31-9), 2% sucrose, 0.5% phytoagar, pH 5.8 (pH adjusted with 1M KOH). The germination trays were placed inside a standard single-well polystyrene plate (VWRI734-2977). Sterile Arabidopsis seeds (cold-stratified in advance for 2–5 days) were individually pipetted into each MYCroplanter such that seeds rested directly on the phytoagar below the MYCroplanter. Plants were allowed to germinate between 5–7 days (see individual experiment methods below for germination days by experiment). A visual depiction of how MYCroplanter parts fit together can be found on our GitHub at https://github.com/mech3132/mycroplanter/blob/main/06_3DPrintingFiles/DIAGRAME_mycroplanters_all_components.pdf.

At 5–7 days old, seedlings were inoculated with bacteria by transferring each MYCroplanter into a 96-plate well pre-filled with 1/2X MS plant growth media and bacterial inoculant. The lids of the 96-well plates were elevated using a 3D printed "lid-brace", which gave plants more vertical space to grow. Plates were sealed with 3M micropore tape and plants were kept in bacterial solution for 7 days. Growth conditions were approximately 110 μMol light in 12h/12h day/night conditions, at 23˚C.

To collect plant health data, MYCroplanters were transferred to a custom 3D printed scanning tray at the end of the experiment, covered in a piece of glass, and inverted onto a Epson Perfection V850 Pro Photo Scanner. Reflective scans in 48-bit colour at a resolution of 600dpi were taken. Images were processed using a custom Python script, which divided each scanned image into an 8x12 array and separated plants from the background using a green and red ratio filter (see 'Image processing and creating a standardized "health score"').

Plant health was scored using a continuous metric derived from pixel analysis of scanned plant images (S2 Fig). In short, we conducted several preliminary screens of isolate effects for five previously characterized *Pseudomonas* strains (N2C3 (pathogen), WCS365 (protective), CHA0 (protective), CH267 (not protective), and Pf5 (intermittently protective)) (S3 Fig) and found that a principal components plot of plant pixel data clustered into three groups: green plants which were "healthy", red plants which were "stressed/sick", and yellow plants which were "dead" (S2A Fig). Pixel information (hue, saturation, value) was transformed into a single continuous variable ("Health Score") (S2B Fig). Since we found that infection by N2C3 was best described using a binary measure of plant health rather than a continuous value, we thresholded ($>400$) all Health Scores into "Healthy" or "Not Healthy" categories. The scoring method and threshold value was validated by human-derived assessments of random plant images (S2C–S2E Fig). Binary plant health status is reported for most results.

### Media preparation

Throughout the text, "plant media" refers to half-strength Murashige & Skoog Basal Medium with vitamins (MS; 2.22g/L; PhytoTech Labs M519) buffered with half-strength 2-(N-

morpholino)ethanesulfonic acid (MES; 0.5g/L) adjusted to pH 5.8 with 1M KOH. For plant germination, solid plant media also included 0.5% phytoagar (BioWorld #40100072–2) and 2% sucrose (w/v). For growth curves, plant media was supplemented with 30mM succinate. Media was sterilized by autoclaving.

Lysogeny broth (LB) for bacterial culture was prepared by autoclaving 10g peptone, 5g yeast extract, and 5g sodium chloride per L MilliQ $H_2O$ and sterilized by autoclaving. When applicable, agar was added for solid media at 15g/L. Tetracycline for maintenance of fluorescent plasmids was added after autoclaving at 40 μg/mL (LB tet40).

## Sterilization of equipment

All MYCroplanter hardware was sterilized using chlorine gas. To create chlorine gas, 300 mL of commercial bleach (6% sodium hypochlorite) and 6 mL of 12M HCl was mixed in a large glass beaker in a 10L sterilite plastic container for at least 1–3 hours (3 hours if fungal contamination was previously present). After sterilization, MYCroplanter hardware was aired out in a Biosafety cabinet for 20 minutes before handling.

## Sterilization and stratification of seeds

Seeds were sterilized using a 70% EtOH + 0.3% $H_2O_2$ solution. Seeds were placed on a clean filter paper disc inside a small petri dish and saturated with ~750μL of sterilizing solution by completely saturating seeds and filter paper. The solution was allowed to evaporate completely in a laminar flow hood, at which point seeds were sterilely transferred to a microcentrifuge tube and suspended in sterile liquid plant media with 2% sucrose and cold stratified in the dark at 4˚C for 2–5 days.

## Image processing and creating a standardized "health score"

Images were processed using a custom Python script, which uses RGB (red-green-blue) colour intensity to separate plant from background. The image is masked using a union of green and red intensity ratio filters (pixel must either have $(G+1)/(R+G+B+1)>0.38$ or $(R+1)/(R+G+B+1)>0.355$ in order to be considered "non-background". Then, images are divided into 8x12 sub-images (96-well plate array). From each subimage, total plant pixel count, median RGB intensities, median G-R, G-B, and R-B pixel differences, and median RGB intensity ratios are extracted.

Plant health scores were calculated using a combination of plant median colour and plant pixel area. We first ran a principal component analysis on RGB, HSV, and HSL colour values of plant images to assess how plant colour varied across 1706 plants and 5 pilot experiments. Plant images were acquired by conducting pilot experiments where plants were inoculated with strains N2C3, WCS365, CHA0, CH267, and Pf5 at different concentrations and combinations on plants of different ages and across 5 temporal experiments (see S3 Fig for experimental designs). We found that plants differed along two major axes: a healthy/sick axis (where healthy plants were large/green and sick plants were small/red), and alive/dead (where living plants were green or red, and dead plants were yellow). We combined these two axis of variation using an ad-hoc formula, (hue*saturation*$\log_{10}$(plant pixel area)) in the HSV colour space (which performed better than an equivalent formula in the RGB colour space) (S2B Fig) to algorithmically rate plants in terms of "plant health".

To validate these health scores as appropriate measures of plant health, we compared the health score algorithm rankings to human-derived rankings of plant health across 84 plants (S2C Fig). We provided 10 random plant images in 10 sets and asked each person to rank plants from most to least healthy; then group plants into either "healthy" or "not healthy"

groups. We found that the HSV algorithm performed better than the RGB algorithm when compared against human-derived rankings. The HSV algorithm performed poorly on plants that were dual-toned: that is, when leaves were half green and half yellow. When these plants are removed from the analysis, the HSV algorithm ranks plants in a similar manner to humans. Since dual-toned plants are relatively rare in our dataset, we concluded that the HSV algorithm was an appropriate measure of plant health in the context of our experimental parameters.

Plants can experience different kinds of stressors (infection, drought stress, suboptimal pH, high bacterial load) and therefore "plant health" is not likely captured by a single continuous metric. Traditionally, plant weight (dry or wet) or leaf area has been used as health metrics because these qualities are easily measurable (compared to differential gene expression, for example). We posit that the health scores we use- while somewhat arbitrary- are no less arbitrary than using "plant weight" as a health metric.

## Bacterial culturing conditions for *Pseudomonas* strains

*Pseudomonas* bacterial strains were grown overnight (~16h) in LB at 28˚C in a shaking incubator. Strains that contained fluorescent plasmids were grown in LB tet40. Overnight cultures were pelleted at 10,000 x g for 1 minute and resuspended in ½ MS ½ MES pH 5.8 plant media. Cultures with antibiotics were spun down a second time and resuspended in plant media to remove residual antibiotics. Cell doses were estimated using a Versamax 96-well plate reader measuring absorbance at 600nm of a 100μL sample (referred to here on out as "Abs600" measurements). Absorbances were converted to OD (absorbance at 600nm of a 1cm cuvette, referred to here on out as "OD600"), which were then used to estimate cell doses assuming 1 mL of OD600 of *Pseudomonas* contains $5x10^8$ cells. Cell doses below are reported in cell abundances (rather than absorbance) to avoid confusion over absorbance versus optical density measurements, and for ease of modelling.

## Estimating cell abundance with fluorescence

To quantify changes in competition outcome between protective (WCS365) and pathogenic (N2C3) strains *in planta*, we engineered WCS365 strains with and without a *lacZ* insertion for blue/white colony counting on X-gal containing plates. We also transformed WCS365 (with and without *lacZ*) and N2C3 with rhizosphere stable plasmids that constitutively express the fluorescent proteins from the Pc promoter on the same plasmid backbone *pSW002-Pc-mNeonGreen* (Neon) or *pSW002-Pc-E2-Crimson* (Crimson) for fluorescence-based population tracking [29]. *LacZ* genomic transposon insertions were described in Wang et al 2022 [19]. Plasmids with Neon and Crimson fluorescent markers were transformed into cells via electroporation. The excitation/emission ranges used to read crimson and neon fluorescence were 612/646 and 495/525 nm, respectively.

In our experiments, we estimated bacterial abundance through fluorescence of liquid in the wells. We removed plants from 96-well arrays before reading fluorescence in a plate reader because plants would both physically (as a result of being too tall to fit in the plate reader) and fluorescently (due to plant auto-fluorescence) interfere with readings. Previous work in larger (48-well) hydroponic Arabidopsis growth systems have shown that bacterial abundances in the liquid surrounding roots is proportional to bacterial abundances on the root itself [10]. This suggests that the primary carbon source for bacteria in a 48-well system is plant root exudate. Since the volumes of liquid we used with MYCroplanters were smaller than the volumes used in previous 48-well hydroponic systems, the most abundant carbon source in the wells

was also likely to be plant root exudate. Thus, we expect bacterial dose in the liquid within each well to be an accurate representation of populations on the roots.

In pilot experiments where we competed Neon and Crimson labelled N2C3 against each other in the rhizosphere, we observed that inconsistent plant water uptake rates and plant root length (sick plants had smaller roots than healthy plants) meant that it was not appropriate to compare absolute cell abundance estimates between treatments. In contrast to cell abundance estimates, $\log_2$-fold ratios were robust across plants (regardless of size, evaporation, etc.) because relative population sizes between non-pathogen and pathogens should remain constant regardless of total cell abundances. Thus, we directly calculated $\log_2$ fold-change ($\log_2$ of ratios) of blanked fluorescence values to estimate relative abundance of the two cell types. A full explanation of this method can be found in the supplemental methods (S1 Methods"), but in short, the $\log_2$ fold change of blanked fluorescence values could be mathematically equated to $\log_2$ fold change of cell types by transforming values with a constant as follows:

Assuming a perfect standard curve (the intercept is (0,0)), the relationship between fluorescence values (F) and OD (OD) can be described by OD = m*F, where m is the slope relating OD to fluorescence values in a standard curve. Therefore, $\log_2$ ratios of OD is equal to:

$$\log_2(OD_{crim}/OD_{neon}) = \log_2(m_{crim}/m_{neon}) + \log_2(F_{crim}/F_{neon})$$

Through this formula, $\log_2$ fold change of OD can be calculated from $\log_2$ fold change of fluorescence values by adding a "slope constant" ($\log_2(m_{crim}/m_{neon})$). This constant was derived by comparing the fraction of colonies positive for lacZ expression on plates with the corresponding blanked fluorescence ratios in a subset of 111 samples (S10 Fig).

## Pilot experiments

We conducted 5 pilot experiments to capture variability in plant healthiness, and also to verify that MYCroplanters could replicate previous findings in the *Pseudomonas*-Arabidopsis system. Plate maps and experimental designs for all experiments are visualized in S3 Fig, but in short: WCS365, CHA0, CH267, Pf5, and N2C3 were inoculated in monoculture and non-pathogen and pathogen pairs on three ages of plants (5, 6, 7 day old seedlings). Concentrations of N2C3 and WCS365 were varied in three out of five of the experiments. The image data collected from these pilots was used to develop the "Health Score" metric (S2 Fig).

## Inoculation ratio experiment

To test whether protection against the pathogen *Pseudomonas fluorescens* N2C3 was ratio-dependent, we inoculated plants with 24 combinations of pathogen and non-pathogen cell dose at three plant ages (experimental design found in S6 Fig). N2C3 was inoculated at four cell abundances: 0, 1E2.6, 1E4.6, and 1E6.6 per 275 µL. The protective strains *P. fluorescens* WCS365, *P. protegens* CHA0, *P. fluorescens* CH267, and *P. protegens* Pf5 were inoculated on plants at 6 cell abundances: 0, 1E2.6, 1E3.6, 1E4.6, 1E5.6, and 1E6.6 per 275µL. For brevity, we round all cell log abundances to the nearest whole number (0,1E3,1E4,1E5,1E6,1E7) in all subsequent references. Plants were inoculated at either 5 days old; 6 days old; or 7 days old (meaning final plant age was either 12, 13, or 14 days old). Each protective strain was combined with N2C3 in every factorial combination of cell dose (4x6), which we refer to as one "set". At least two sets of each protective strain were included in each temporal "experiment". Sets for each non-pathogenic bacterial strain were replicated in two temporal experiments. WCS365 and Pf5 sets were replicated in three separate experiments (2023-06-14; 2023-07-26; 2023-08-02) while CHA0 and CH267 sets were replicated in two separate experiments (2023-07-26; 2023-08-02). Monoculture effects of non-pathogens were replicated in two additional experiments

(2023-06-21; 2023-06-27) to better resolve monoculture effects. For the experiments on 2023-06-14, 2023-06-21, and 2023-07-26, 5-, 6-, and 7-day old plants were germinated at staggered times but inoculated on the same day with the same overnight cultures. For the experiments on 2023-06-27 and 2023-08-02, plants were germinated on the same day but inoculated either 5, 6, or 7 days later, which meant different overnight cultures were used for each day.

To determine whether initial cell number at inoculation affected final bacterial load on plant roots, we conducted an additional experiment where WCS365 and N2C3 were inoculated at varying concentrations (6 10x serial dilutions starting at 1E7.6 cells) and final absorbance (600nm) of each well as measured after 7 days of growth on plants (experiment 2023-05-24_age_concentration). One column (column6) was removed due to higher than expected absorbances in controls, suggesting contamination of that column. Plate setup can be found in (S5B Fig) and results are shown in (S5A Fig).

## Plant mutant experiment

The plant immune mutants *bbc* (*bak1*, *bkk1*, *cerk1*) and *bik1* were inoculated with WCS365 and N2C3 to determine whether the ability of WCS365 to protect against N2C3 changed when plants were immunodeficient. We inoculated N2C3 at four abundances (0, 1E3, 1E5, 1E7 cells). Protective strains (WCS365, CHA0, and Pf5) were inoculated at two concentrations (0, 1E5). Different inoculation ratios were used in order to find the optimal ratio that maximized differences in protection (if any). Plants were five days old at inoculation. We looked for the minimum non-pathogen to pathogen ratio that showed loss of protective ability in any plant mutant relative to wildtype and the maximum non-pathogen to pathogen ratio that showed gain of protective ability in any plant mutant relative to wildtype. Experimental design can be found in S9 Fig.

## Community dominance with fluorescence experiment

Ratio experiments for WCS365 and N2C3 were replicated with fluorescently labelled strains on 5-day-old plants. Replicate experiments were completed three times (2023-09-13, 2023-09-14, 2023-11-28), where each experiment had two plates of N2C3-neon/WCS365-crim and two plates of WCS365-neon/N2C3-crim and used different plant germination and bacterial overnight batches. At the end of the experiments, plates were read for fluorescence.

For experiments 2023-09-13 and 2023-09-14, WCS365 strains (both Neon and Crimson) contained a *lacZ* genomic insertion so that cell counts could be mapped back to neon and crimson fluorescence values. The fraction of blue colonies formed on X-gal plates was quantified for 87 wells (S10 Fig). These wells were distributed across four plates and included inoculation densities from zero to 1E7 cells for both WCS365 and N2C3. According to our calculations above (see Estimating cell abundance with fluorescence), the slope between $\log_2$ ratios of fluorescence and CFU counts should be approximately 1. Therefore, we removed extreme outliers (3 wells) in our CFU count subset and fit a linear model, forcing the slope to equal 1, to obtain a "transformation constant" that allowed us to correlate fluorescence-based log-ratio cell abundances to empirical CFU counts. This constant was used to transform all fluorescence data.

## Estimating plasmid loss

Plasmid maintenance was quantified for one-week-old cultures by plating colonies on LB plates, and then replica plating them on selective (LB tet40) plates to differentiate plasmid-positive and plasmid-negative colonies. Plasmid maintenance was measured over two experimental replicates. One set was sampled from set three (2023-11-28) of the fluorescence community dominance experiment, and another was set up de novo with the same parameters. In short, 5-day-old plants

were inoculated with approximately 100,000 cells (sampled from experiment 2023-11-28) or 10,000,000 cells (de novo experiment) of one of six strains (WCS365-crimson, WCS365-neon, N2C3-crimson, N2C3-neon, and plasmid free WCS365 and N2C3) and incubated for 1 week. Liquid from 2–3 wells was streaked onto LB agar plates. From these plates, approximately 100 isolated colonies were picked randomly and dotted onto LB tet40 plates µg/mL. The proportion of colonies that grew on the LB tet40 after overnight incubation at 28˚C were recorded. To ensure that there was not a bias in initial plasmid maintenance in the overnight cultures and that the process of transferring colonies from LB to LB tet40 did not induce plasmid loss, the original inoculum (that was used to inoculate plants) was also plated at the time of inoculation in serial dilutions onto LB agar, from which approximately 50 random colonies were picked and transferred to LB tet40 to assess plasmid loss in the inoculum.

As plants were grown with bacteria for a full week, we wondered whether fluorescent plasmid loss would bias our results due to inaccurate strain reporting. We quantified plasmid loss from all four fluorescent strains (WCS365-crimson, WCS365-neon, N2C3-crimson, N2C3-neon) and found that N2C3 strains lost their plasmid more often than WCS365 (S15 Fig). For the N2C3-neon strain, as few as half of the colonies retained the mNeon plasmid after 1 week of incubation on plants, while WCS365 strains kept their plasmids in over 90% of colonies. These proportions were consistent across two replicate tests, which included different inoculation concentrations (1E5 cells and 1E7 cells). We found that plasmid loss can be incorporated into our strain abundance estimates by using a conversion constant derived specifically from 1-week incubations. Given that plasmid loss is consistent across two one-week trials, and that fluorescence-independent methods of cell quantification confirm the accuracy of fluorescent ratios, we believe the method of using $\log_2$-ratio transformations produces reasonable estimates of relative strain abundance.

## Priority effects experiment

We performed priority effect experiments using two methods. In the first method, ("well-to-well" method), plants were inoculated at 5 days old with either N2C3 of WCS365 first, followed by the second strain immediately (3 seconds, referred to as "dip"), 3 hours, 6 hours, 24 hours, or 48 hours later. Inoculation of the first strain was accomplished by placing seedlings roots in a hydroponic solution of the first strain (suspended in 1/2MS 1/2MES) in a 96-well plate. Seedlings were then removed from the first well and placed in a second well (without rinsing) containing the second strain (suspended in 1/2MS 1/2MES). Plants were incubated in the second-strain-containing well for the remainder of the experiment. This method was repeated four times, where the first set included only 24 and 48h time points.

A second method ("double dip" method) was used to ensure that cell abundances for the first and second strains were approximately equal when plants were only "dipped" in the first strain. Plants were dipped in the first strain for 3 seconds, dipped in the second strain for 3 seconds, and placed in a well full of sterile plant media. This method was repeated twice. For both "well-to-well" and "double-dip" methods, strains were diluted using plant growth media to cell numbers of approximately 0.001 Abs600.

Finally, we conducted an experiment to assess the relative importance of priority effects versus cell abundance on community assembly ("priority versus ratio" experiment). Plants were incubated in strain 1 (Abs600 = 0.001) for 6h before being transferred to a well containing different concentrations of strain 1 (Abs600 = 0.0001, 0.001, 0.01, 0.03). One set of the experiment overlapped with the 4th set of "well-to-well" experiments; whereas one set was generated de novo.

The relative abundance of CFUs on plant roots versus the well were quantified in a subset of all priority effect experiments. To determine how cell abundance on the root increased over

time, random plants at 3h, 6h and 24h timepoints from the "well-to-well" experiment were sampled. Additionally, the number of CFUs on plant roots at the 6h time point from "priority versus ratio" experiments were also quantified. Lastly the number of strain 2 CFUs in wells containing Abs600 cultures at 0.0001, 0.001, 0.01, and 0.03 (into which 6h plants were transferred) were quantified alongside roots. To sample root CFUs, seedlings were removed from the first well and transferred to a tube of 500 µl 1/2MS 1/2MES. One seedling was added per tube and three seedlings were transferred in total per strain. Two sterile metal beads were added per tube and plants were homogenized using the Qiagen TissueLyser II for 1 min at 30 Hz. Seedling homogenate (strain 1) and cell cultures from wells (strain 2) were serially diluted, spotted in 5µL dots on LB plates in triplicate, and CFUs were quantified. Experimental design for all priority effect experiments can be found in (S11 Fig).

We also compared the strength of priority effects *in vivo* and *in vitro* using parallel experiments in a solid agar system. Arabidopsis seeds were sterilized using chlorine gas (300mL 6% bleach + 6mL HCl) for one hour and cold-stratified in water for 2 days. Seeds were germinated on vertical 1% phytoagar plates with 1/2MS 1/2MES 2% sucrose for 6 days. Seedlings were transferred to 1/2MS 1/2MES (no sucrose) 1% phytoagar plates, and allowed to acclimatize for one day before inoculations. For concurrent inoculations, 6µL total of 0.001OD cultures were added to the plant (WCS365 and N2C3 were mixed at 1:1). For delayed inoculations, 6µL of 0.001 OD culture was inoculated on the first day, and 6µL of 0.001OD culture was inoculated on the second day. For *in vitro* treatments, cultures were spotted onto identical phytoagar plates with no plants growing. In each treatment, one of the strains included a *lacZ* genomic insertion. Relative abundances of CFUs were counted as per methods below (*LacZ* plate counting).

## LacZ plate counting

LB agar plates with 20 ug/mL X-gal were used to distinguish strains with *lacZ* insertions (blue), and strains without *lacZ* insertions (white). Previously, the *lacZ* insertions were shown to have no fitness defects relative to wildtype[25]. When samples were taken from MYCroplanters, cells were sampled from the hydroponic liquid, and serially diluted using plant media. When samples were taken from plants grown on solid phytoagar, whole plants were placed in 1.5mL microcentrifuge tubes with a metal bead and 500µL 10µM $MgSO_4$ and shaken at 28 hz for 10 minutes. For samples taken from *in vitro* experiments, 10µL of 10µM $MgSO_4$ was pipetted up and down on the plate where bacteria were spotted, and transferred to 490µL of $MgSO_4$ in a 1.5mL microcentrifuge tube. Cell dilutions were spotted in 10µL dots on X-gal plates in duplicate or triplicate (technical replicates). In cases where duplicates were not similar in counts, the sample was discarded. Previous work has shown that cells found in the hydroponic liquid mirror communities found directly growing on Arabidopsis plant roots.

## Bacterial growth curves

Bacterial cultures were grown in clear flat-bottom 96-well plates for all growth curves. Overnight cultures were washed and prepared to Abs600 = 0.01 with a final volume of 200µl. Growth was monitored on a shaking plate reader (Molecular Devices VersaMax or SpectraMax microplate reader) set to 28˚C. Readings were taken once every 15 min or 20 min (depending on the machine used) for 24 hours. The growth curves for strains grown in LB were repeated twice (3–4 technical replicates per strain for each trial). Growth curves for strains grown in ½MS½MES + 30 mM succinate were repeated twice (9–12 technical replicates per strain for each trial) except for N2C3 and WC365 without plasmids or *lacZ* insertion, which are from a single trial (11–12 technical replicates per strain).

## Statistical analysis

All statistical analysis was conducted in R (4.3.2)[45]. Data manipulation and plotting were accomplished using the tidyverse [46], lubridate [47], cowplot [48], gridExtra [49], plotwidgets [50], and ggbiplot [51] packages. Bayesian regression was performed using the 'brms' package [52–54]. PCAs were created with the 'prcomp' function in the MASS package [55].

To test the effect of inoculation concentration and ratios on the likelihood of plants becoming infected, we used a "Bernoulli" model where both cell dose ($\log_{10}$ cells of pathogens and non-pathogens) and their interaction (which would identify ratio-dependent effects) were included as predictors. Since cell dose and inoculation ratios are correlated, the initial full model would randomly attribute variation to one of those two predictors. To remedy this, we first ran a "monoculture only" model that included only treatments with one strain (N2C3 only, WCS365 only, CHA0 only, CH267 only, Pf5 only, no bacteria). The mean and standard deviations of posterior draws for the intercept and cell dose of each strain were included as informative priors in the full model. Additionally, in the full model, we set the interactions between MOCK treatments and cell dose to a constant of zero, since there is no difference in cell dose between any MOCK treatments. The final model included plant age, pathogen cell abundance, non-pathogen cell abundance, and the interaction between pathogen cell abundance and non-pathogen cell abundance within each non-pathogenic strain. Experimental plate was included as a random effect. Healthy/not healthy (health score > or < 400) was the response variable. Iterations were increased to 4000. Posterior draws were summarized by their mean value, 95% credible intervals, and standard deviations. Leave-one-out comparisons using loo() (from the 'brms' package) were used to compare three different competition models: (1) a model with only cell concentrations, (2) a model with cell concentrations and their interaction term, and (3) a model with only the interaction term between cell concentrations.

Using the same data as the inoculation ratio experiment, we also asked whether plant age interacted with any treatments. We ran a Bernoulli Bayesian model with plant health (healthy/ not healthy) as a response to plant age, plus the interaction between plant age and pathogen treatment, non-pathogenic bacteria treatment, or both. Iterations were increased to 4000 to keep the model consistent with others.

To test the effects of pathogen/non-pathogen co-inoculation on plant immune mutants, plant mutant experimental data was split into two sets: one set included all treatments with WCS365, CHA0, and Pf5 monoculture treatments only, and one set with each strain co-inoculated with N2C3. A Bayesian model was used to predict effects of plant genotype (Col-0, *bbc*, *bik1*) on N2C3 + non-pathogen treatments, using Col-0 as the baseline. We also ran a second Bayesian model with strain monocultures only to determine whether the strains alone affected plant health differently for each plant mutant. For each model, the effects of plant mutant and pathogen exposure (with interaction) was determined using a Bernoulli model with uninformative priors and 4000 iterations to improve model convergence. Max tree depth was increased to 15 to resolve convergence issues. We calculated 90 and 95% prediction intervals using brms (predict_linpred).

For the community dominance fluorescence experiment, we conducted model comparisons between models that used fluorescence ratios, inoculations, or both to predict probability of plant healthiness. Model comparisons were done using LOO.brmsfit(), which uses Bayesian leave-one-out estimates of the expected log pointwise predictive dose and is similar to wAIC comparisons. The best model (which used only fluorescence and total cell dose, and not inoculation ratios, as predictors for plant healthiness) was used to generate 95% prediction intervals using posterior_linpred().

To determine whether priority effects were responsible for which strain dominated the community, we used data from"well-to-well" experiments to ask whether WCS365:N2C3 $\log_2$ fold fluorescence, transfer time, or order of inoculation predicted plant health in a Bayesian Bernoulli model. We compared models with and without $\log_2$ fold change fluorescence and order of inoculation+delay predictors to determine which was a better predictor of plant health. Models were compared using LOO.brmsfit().

We then asked whether priority effects during early colonization predicted community dominance in "double-dip" experiments. We modelled WCS365:N2C3 $\log_2$ fold change fluorescence as a response to inoculation order and inoculation CFU ratios using a Bayesian model (iterations = 4000). Using the same data, we also asked whether plant health was, in turn, predicted by inoculation order or WCS365:N2C3 $\log_2$ fold change fluorescence values (Bernoulli model, iterations = 4000).

Finally, we asked whether altering the cell abundance of strain 2 could overwhelm priority effects. We used the "priority versus ratio" experiment data to model whether strain dominance (WCS365:N2C3 $\log_2$ Fold Change) at the end of experiment was predicted by inoculation order or cell ratios at inoculation. We used a Gaussian Bayesian model with iterations 4000.

In all experiments, raw data was visually inspected for contaminated negative controls. Wells suspected of being contaminated were removed, for example if a well with only Crimson-labelled cells had unexpectedly high Neon fluorescence values.

## Dryad DOI

10.5061/dryad.w9ghx3fxd [56].

## Supporting information

**S1 Methods. A detailed description and justification of $\log_2$-fold ratio calculations for determining ratios between strains with two fluorescent protein markers.**
(PDF)

**S1 Fig. Platemap for N2C3 vs. WCS365 pilot experiment to validate MYCroplanter sensitivity in detecting our model system effects.** Each panel represents one 96-well plate. Mock treatments were plain 1/2MS 1/2MES pH5.8 plant growth media.
(PDF)

**S2 Fig. Pixel data from scanned plant images reveals a multi-model distribution of plant health scores that align with human-derived rankings of plant health.** (a) Principal components analysis of median pixel values using the HSV (Hue, Saturation, Value) colour model. Plants are separated by two primary axes: light green/dark green(healthy/stressed) and green/yellow (alive/dead). (b) HSV pixel values were transformed into a single continuous health score metric (hue*saturation*$\log_{10}$(all_plant_pixels+1)*1000), which reflects the roughly tri-modal distribution of plant health states (healthy / stressed / dead). (c) Algorithm-derived "Health Score" metrics using HSV and RGB values were compared to human-derived ranks. Human rankers ranged from lay-people (no scientific background) to expert (PhD). Humans and scoring algorithms were asked to rank 10 sets of 10 plants (85 total plant images; some images were repeated between sets for validation purposes) in order from "least healthy" to "most healthy". Each was also asked to divide plants into "not healthy" and "healthy" groups. We found that three plants were consistently ranked differently between humans and our algorithm (7714, 3443, 5158), and that these plants were two-toned (half green half yellow). This caused our algorithm results to perform poorly. When these plants were removed, our Health Score algorithm performed comparably to most human rankers. (d) Heatmap of paired

Kendall correlations (tau) between each human and/or algorithm plant ranks. (e) "Healthy" / "Not healthy" classifications by humans were compared to algorithm classifications at different Health Score thresholds. To avoid over-interpretation of health score values, we categorized plants as "healthy" or "not healthy" based at a threshold of 400. Binary response variables are used and reported in the main results and text.
(PDF)

**S3 Fig. Plate map and experimental design for pilot data used to derive plant health scores.** Plants were inoculated at three different ages (5, 6, 7 days) with four different concentrations of pathogen and 4 non-pathogenic bacterial strains across 5 experiments. The breadth of experimental variables used was to increase the range of plant health observed in our pilot data.
(PDF)

**S4 Fig. Effect of strain presence and dose for *Pseudomonas* strains in monoculture on probability of plants remaining healthy.** Points are median estimates from a Bayesian Bernoulli model, thick bars are 90% credible intervals, and thin bars are 95% credible intervals.
(PNG)

**S5 Fig. Final absorbance of cells in 96-well plates after 1 week of incubation with plants shows that carrying capacity is similar regardless of initial inoculation concentration.** (b) Platemap and experimental design for data shown in (a). Plants were inoculated with different concentrations of pathogen and protective. Cell abundance units are on the $\log_{10}$ scale $(1 = 10^1)$.
(PDF)

**S6 Fig. Plate map and experimental design for experiment testing effect of inoculation ratio and plant age on plant health.** Plants were inoculated at three different ages (5, 6, 7 days) with 4 and 5 concentrations of pathogen and protective, respectively, across 5 experiments.
(PDF)

**S7 Fig. Effect of non-pathogen and pathogen dose across bacterial strain on the probability of plants remaining healthy.** Points are median estimates from a Bayesian Bernoulli model, thick bars are 90% credible intervals, and thin bars are 95% credible intervals.
(PNG)

**S8 Fig. Older plants are less susceptible to infection and benefit more from protective symbionts than younger plants.** (a) Coefficient estimates for different variables associated with plant age show that the effect of plant age interacts with bacterial strain. In the first panel, the coefficient for the independent (global) effect of plant age is positive, meaning older plants are more likely to remain healthy, overall. In the second panel, we show the interaction between age and each monoculture treatment. Negative coefficient numbers mean there is a weaker effect of age when treated with that strain, relative to the global positive effect of plant age. Here, a negative value for N2C3:plant age means that while plants are usually healthier with age, the effect of plant age when inoculated with N2C3 is less different between 5-day-old and 7-day-old plants (ie 7-day-old plants are not as healthy as one would predict if there was no interaction between N2C3 and plant age). In the third panel, we show the interaction of plant age and each N2C3 + non-pathogenic bacterial strain treatment. When coefficients are positive, it means older plants benefit more from protection of the non-pathogenic strain, relative to what might be expected due to plant age alone. (b) Using the model from (a), we generated prediction intervals for the probability of healthy plants. Dots represent single plants. Lines

represent median posterior predictions according to the Bayesian Bernoulli model; ribbons represent 95% prediction intervals.
(PDF)

**S9 Fig. Plate map and experimental design for plant genotype experiment.** Three plant genotypes (Col-0, *bbc*, and *bik1*) were inoculated with WCS365, CHA0, and Pf5 with or without N2C3 at three different protective:pathogen ratios. In our paper, we present only 1:1 and 1–0.1 protective:pathogen treatments because plants from 1:10 treatments were overwhelmingly "not healthy", which means we could not observe any differences between control (Col-0) and treatment (*bbc*, *bik1*) plant genotypes.
(PDF)

**S10 Fig. Plate map and experimental design for data shown in fluorescence experiment (Fig 4).** Strains were labelled with either Crimson or Neon plasmids. Black outlined circles indicate wells that were used for serial dilution CFU counting, in order to get *lacZ*-based estimates of each strain's cell density, which was fluorescence-independent. Top labels for each panel refers to the temporal experiment, whereas bottom labels for each panel refers to different replicate plates within each experiment.
(PDF)

**S11 Fig. Plate map and experimental design for all priority effects experiment in Figs 5 and 6.** Top labels for panels indicate which plate replicate each panel represents, whereas horizontal rows indicate which temporal experiment each plate belongs to. A time delay of 0.01 refers to "seconds" apart in our dip treatments. A time delay of zero indicates simultaneous inoculation.
(PDF)

**S12 Fig. Cell densities of WCS365 and N2C3 on plant roots in monoculture across time.** N2C3 reaches slightly higher cell densities than WCS365 on plant roots when in monoculture.
(PNG)

**S13 Fig. Strong priority effects exist between WCS365 and N2C3 when grown *in planta* but not *in vitro*.** Shown are the proportions of WCS365 and N2C3 colonies on solid plant agar when grown with and without plants under different inoculation regimes. Plants were inoculated with WCS365 or N2C3 either at the same time (Concurrent inoculation) or with a 24 hour delay (Delayed inoculation).
(PDF)

**S14 Fig. Growth curves of all strains used in experiments.** WCS365 and N2C3 do not differ in growth rate or carrying capacity in LB media, with or without plasmids. There is a slight burden on N2C3 with both m-Crimson and m-Neon plasmids in minimal media.
(PDF)

**S15 Fig. Plasmid loss for WCS365 and N2C3 strains with Neon and Crimson plasmids.** N2C3 strains lost both plasmids at higher rates than WCS365, and this effect was particularly strong for N2C3-Neon.
(PDF)

## Acknowledgments

The fluorescent plasmids; pSW002-Pc-mNeonGreen & pSW002-Pc-E2-Crimson were a gift from Rosemarie Wilton (Addgene plasmid # 205016; http://n2t.net/addgene:205016; RRID: Addgene_205016 & # 111252; http://n2t.net/addgene:111252; RRID:Addgene_111252).

## Author Contributions

**Conceptualization:** Melissa Y. Chen, Leah M. Fulton, Quentin Geissmann, Cara H. Haney.

**Data curation:** Melissa Y. Chen, Leah M. Fulton, Ivie Huang.

**Formal analysis:** Melissa Y. Chen, Leah M. Fulton.

**Funding acquisition:** Melissa Y. Chen, Leah M. Fulton, Aileen Liman, Cara H. Haney.

**Investigation:** Leah M. Fulton, Ivie Huang, Aileen Liman, Cara H. Haney.

**Methodology:** Melissa Y. Chen, Leah M. Fulton, Ivie Huang, Aileen Liman, Sarzana S. Hossain, Corri D. Hamilton, Siyu Song, Quentin Geissmann.

**Project administration:** Melissa Y. Chen, Cara H. Haney.

**Resources:** Cara H. Haney.

**Supervision:** Kayla C. King, Cara H. Haney.

**Validation:** Leah M. Fulton, Corri D. Hamilton.

**Visualization:** Melissa Y. Chen, Leah M. Fulton.

**Writing – original draft:** Melissa Y. Chen, Leah M. Fulton.

**Writing – review & editing:** Melissa Y. Chen, Leah M. Fulton, Sarzana S. Hossain, Quentin Geissmann, Kayla C. King, Cara H. Haney.

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
