## [Decision Letter · Decision Letter 0]

1 Dec 2024

PPATHOGENS-D-24-01866

Order among chaos: high throughput MYCroplanters can distinguish interacting drivers of host infection in a highly stochastic system

PLOS Pathogens

Dear Dr. Chen,

Thank you for submitting your manuscript to PLOS Pathogens. After careful consideration, we feel that it has merit but does not fully meet PLOS Pathogens's publication criteria as it currently stands. Therefore, we invite you to submit a revised version of the manuscript that addresses the points raised during the review process.

Please submit your revised manuscript within 60 days Jan 30 2025 11:59PM. If you will need more time than this to complete your revisions, please reply to this message or contact the journal office at plospathogens@plos.org. Please include the following items when submitting your revised manuscript:

We look forward to receiving your revised manuscript.

Kind regards,

Nian Wang

Academic Editor

PLOS Pathogens

Shou-Wei Ding

Section Editor

PLOS Pathogens

Michael Malim

Editor-in-Chief

PLOS Pathogens

orcid.org/0000-0002-7699-2064

**Additional Editor Comments:**

Apologies for the delay. Four experts agreed to review this manuscript. However, two reviewers could not submit their reviews timely, but did cause some delay in the decision making process. The two experts who reviewed the manuscript are very positive about this manuscript. Since no experiments are required, the manuscript seems to warrant a recommendation of minor revision. However, reviewer 2 had raised some very critical points that must be addressed by the authors related to microbiome intention with a 2-species approach. While the reviewers and the editors are convinced that this device would be very useful for complex microbiome interactions, the current test is the first step towards testing its full potential in microbiome study. So the logic needs to be thoroughly revised to better justify its potential and the current experimental approach.

**Journal Requirements:**

At this stage, the following Authors/Authors require contributions: Leah M Fulton, Ivie Huang, Aileen Liman, Sarzana Hossain, Corri Hamilton, Siyu Song, Quentin Geissmann, and Kayla C King. Please ensure that the full contributions of each author are acknowledged in the "Add/Edit/Remove Authors" section of our submission form.

3) We noticed that you used the phrase 'data not shown' in the manuscript. We do not allow these references, as the PLOS data access policy requires that all data be either published with the manuscript or made available in a publicly accessible database. Please amend the supplementary material to include the referenced data or remove the references.

Potential Copyright Issues:

i) Figures 1A, and S15. Please confirm whether you drew the images / clip-art within the figure panels by hand. If you did not draw the images, please provide (a) a link to the source of the images or icons and their license / terms of use; or (b) written permission from the copyright holder to publish the images or icons under our CC BY 4.0 license. Alternatively, you may replace the images with open source alternatives. See these open source resources you may use to replace images / clip-art:

5) Thank you for stating that 'FOR PURPOSES OF PEER REVIEW, PLEASE ACCESS DATA THROUGH THIS LINK: https://datadryad.org/stash/share/pB9zBT5xRrKy9-3aAS92eVjgwn1MF2qpHVU82Ow_Ncg Processed data.' We couldn't access the dataset through the provided link. Please provide us with a direct link to access the dataset. If your manuscript is accepted for publication, you will be asked to provide these details on a very short timeline. We therefore suggest that you provide this information now, though we will not hold up the peer review process if you are unable.

**Reviewers' Comments:**

Reviewer's Responses to Questions

**Part I - Summary**

Reviewer #1: In this manuscript, the authors describe a new platform to increase the throughput of plant-microbe interactions. Further, they use previously characterized interactions to demonstrate a proof of concept of the system to generate robust results, as well as provide image processing pipeline to assign a binary plant health outcome. The high throughput results generated allowed the authors to quantify differences between protective strains against N2C3 infection to investigate a number of variables such as inoculation ratio, timing on inoculation and impact of plant genotype on plant health outcome, despite the biological diversity of plant interactions with multiple microbes. The authors demonstrate that such technical advancements hold the potential to reveal important details in the mechanisms of protection conferred by previously characterized commensal strains.

Reviewer #2: “Order among chaos: high throughput MYCroplanters can distinguish interacting drivers of host infection in a highly stochastic system,” by Chen et al. describes the novel tool MYCroplanters to conduct simultaneous and sequential strain inoculation experiments in Arabidopsis thaliana. Because their high-throughput tool facilitates experiments, the authors are able to present a significant amount of data demonstrating plant health outcome effects and final occurrence ratio of two inoculated strains, one of which is always the same pathogen strain, and the other of which is one of a variable set of closely related strains. They find that the identity of the paired strain, the ratio at which the paired and pathogen strain are inoculated, and the genotype of the plant alter health outcomes. They also find that the order and manner of arrival of the strains alters final 2-strain relative abundance outcomes, and that plant health outcomes and these 2-strain relative abundance outcomes are linked. In sum, the authors have presented a useful new tool to the community, that also is potentially scale-able or transferrable to other host organisms. I very much enjoyed this paper. I think the method is novel and exciting, and the results are useful in their own right.

My critiques are almost exclusively about framing. I think the framing needs to move away from “microbiomes” and community properties as these are only 2-strain communities, and towards epidemiology – as this is a pathogens journal.

**Part II – Major Issues: Key Experiments Required for Acceptance**

Reviewer #1: Because there is such an impressive number of variables tested, the text could provide a bit more clarity to help the reader follow all what the discussion of each presented experiment:

1) It might be easier to follow the text describing the different microbiota and plant genotype variables if "strains" was used to describe the different Pseudomonas strains and "genotypes" was just used to describe the Arabidopsis genotypes.

2) In Figure 3, the orange and yellow data under the "No microbiota" heading are a bit confusing because the ratio of 1:1 and 1:0.01 are really 0:1 and 0:0.01. Perhaps this would be more clear if there was a different key and/or color code from just the "No microbiota" graphs.

3) Figure 5b have pretty extensive label names, a little schematic added to the top or bottom of this graph might make these different conditions more intuitive.

Reviewer #2: First, the abstract and 2nd paragraph of the introduction led me to expect that this manuscript was about manipulating lots of diverse microbiome communities, yet, that is not what is happening. Referring to these 2-strain mixes as “microbiomes” is confusing -- I also hesitate to call them “communities.” Using “protective strains” or “commensal strains” or “non-pathogen strains” or even “microbes” or similar instead of “microbiota” and “microbiome” in the abstract and elsewhere would be more transparent. Similarly, “community structure” and “community dominance” don’t make much sense in a 2-species context, as there is simply relative dominance of one strain versus the other. Again similarly, the fact that final relative abundance of a pathogen and non-pathogen in a 2-strain community and host health are correlated does not seem particularly surprising. Framing the 2-strains as a “community” with “composition” that predicts “infection outcome” seems to inflate the importance too much. However, I *am* completely convinced that this device would be very useful for complex microbiome interactions, and I appreciate the text pointing that out.

Second, and this is key given the journal, I think the authors should add motivation and interpretation of this study in light of epidemiology and disease (theory/modelling or etc.). Intra-host competition strains seems very related to the design and results, maybe that would be a good framework.

Third, given that MYCroplanters are one of the key items presented in the manuscript, I think it is important to provide more discussion about these and their utility. I’m glad to already see a blueprint-style figure & the 3-D printing files. In the supplemental figures, it looks like most treatments had to be applied in spatially grouped ways -- presumably because it is difficult to prevent microbial cross-contamination between wells. I think a (brief!) discussion of the trade-offs in experiment size and the ability to randomize, or the statistical tools used to deal with the blocking/plate/edge effects that might be necessary is warranted.

**Part III – Minor Issues: Editorial and Data Presentation Modifications**

Reviewer #1: There are also a couple of labeling suggestions:

1) Figure 3 x-axis labels should be italicized for the genotypes, with the WT being labeled "Col-0" as it is in the text.

2) In Figure 5a, the times besides the dips have units. The authors provide details on what "dip" means in the text, but perhaps for consistency the "dips" could be relabeled times with unit, such as "3 seconds" or "< 3 seconds".

Reviewer #2: The side panels on figure 4 are a clever and intuitive way to deal with lack of detection. However, why is the x-axis arranged as it is? It would make more sense in numerical order

(100000-1, 100-1, 1-1, 1-100, 1-00000) especially because the results are interpreted that way.

The figures all use the “probability of plant health” or similar, but the manuscript also uses the clearly related “health outcome” as well as less clearly related “infection rate” and “infection outcome”. I think reducing the number of different phrases would be helpful.

Sometimes “dose” is refers to total amount, and sometimes to ratio. Since the results depend on the type, it might be best to use always “total dose” or “dose ratio”. E.g. L303 I think the significant result was for ratio not total dose?

Lines 109-110 in the abstract. I’m uncomfortable with community structure framed as a predictor in the same way as the manipulated variables. The 2-species composition outcomes were also observed, and therefore, we do not know the nature of the relationships between the final composition and plant health. Granted, we have a strong prior here, but it could also be the case that declining plant health allows spikes in pathogen numbers.

Line 153: Commensal is usually reserved for organisms with truly no effect on the host. It seems these strains may be better described as conditionally mutualistic?

Lines 192-196: These summarize content already introduced and could be skipped.

Line 243: I suggest elaborating on “distinct engineering strategies” or skipping

Line 248: I could not easily figure out which of the experiments these data came from.

Line 408: To me “confounded” communicates that priority effects are artificially inflated. Comparing a with b, it looks like the dip + move to the second well with many microbes “eroded” priority effects.

I don’t understand the separation of priority and inoculation effects. I do think the analytical separation makes sense, and that there is a neat result here, but I’m not sure the words to describe it in lines 419-433 make sense. Isn’t getting early access to host nutrients and therefore increasing in population part of what a priority effect is? Also “physcially precluding” the first strain seems to fit more with the “neutral assembly” process because physically precluding would still be just a numbers game? Maybe the early microbes alter how the plant develops & therefore fundamentally alter the habitat (i.e. like ecosystem engineering??). I think I’m missing something the authors could easily explain better.

Lines 494-503: in 7b, the relationships all overlap, which fits with this conclusion. However, in 7a, the relationship between final ratio and plant infection differ depending on the treatment. It seems then that the conclusion should be that inoculation ratio shifts (but only slightly) the nature of how strain ratio predicts infection.

Line 706: Is this missing the absorbance value?

L915: probably seeds not seedlings were made axenic with Cl gas?

PLOS authors have the option to publish the peer review history of their article (what does this mean?). If published, this will include your full peer review and any attached files.

Reviewer #1: No

Reviewer #2: No

**Figure resubmission:**
---

## [Decision Letter · Decision Letter 1]

8 Jan 2025

Dear Dr. Chen,

We are pleased to inform you that your manuscript 'Order among chaos: high throughput MYCroplanters can distinguish interacting drivers of host infection in a highly stochastic system' has been provisionally accepted for publication in PLOS Pathogens.

Best regards,

Nian Wang

Academic Editor

PLOS Pathogens

Shou-Wei Ding

Section Editor

PLOS Pathogens

Sumita Bhaduri-McIntosh

Editor-in-Chief

PLOS Pathogens

orcid.org/0000-0003-2946-9497

Michael Malim

Editor-in-Chief

PLOS Pathogens

orcid.org/0000-0002-7699-2064

Reviewer Comments (if any, and for reference):

Reviewer's Responses to Questions

**Part I - Summary**

Reviewer #1: The authors have very nicely incorporated the suggestions made by both reviewers.

Reviewer #2: I was the previous Reviewer 2. I remain very excited about this manuscript! – for all the reasons I previously mentioned. It has improved substantially as well.

**Part II – Major Issues: Key Experiments Required for Acceptance**

Reviewer #1: (No Response)

Reviewer #2: None

**Part III – Minor Issues: Editorial and Data Presentation Modifications**

Reviewer #1: (No Response)

Reviewer #2: I have two very minor comments, and I noticed some typos.

1. This is about my former minor comment about the former Lines 109-110 in the abstract, and related section. I think I was unclear. The minor section “Regardless of inoculation conditions, the likelihood of plants developing disease is predicted by non-pathogen to pathogen ratio in the rhizosphere” should not infer causation in lines 497-498. I also suggest using “correlates with” instead of “predicts” or “is predicted by” in this section. Because disease and final strain ratio outcomes are both measured, not manipulated, it is not appropriate to infer causation. Otherwise, this section (Lines 478 - 498) and related text elsewhere is great

2. In relation to my third comment from the previous round. As Fig S6 shows (and for some other experiments), the nature of the blocking means some treatments are more likely to contain edge effects than others. However, given the repetition of many of the treatments across experiments, and the consistency of the results, I’m not concerned with the authors’ conclusions based on these experiments. Yet, I like the authors’ last sentence in response to my comment, and I soft suggest something similar be in the manuscript somewhere (anywhere!). For example: “While our experiments used blocked designs, we note that the setup is compatible with liquid handling robots, which makes full randomization possible.” If the authors do not wish to include, that is fine.

Typos

L26 I suggest exchaning “predictor” for “correlate”

L77 “have” should be “has”

L131 should “resulted in” be “resulted from” or “quantified” or? -since health score didn’t cause the phenotypic difference per se

Fig 1 & 2 caption and line 221: The cells/inoculation units should be spelled out so we don’t have to scroll to the methods. i.e. is it log10 of total cells inoculated into the cell or of cells per uL or?

Fig S2 panels d, e, missing explanations in the caption.

Caption Fig S3 “bread” is meant to be “breadth” I think.

Figure S5: Are the x-axis units log10 based as elsewhere? i.e. “7” for “cells” does not actually mean 7 cells expected to have been inoculated?

The left edge of Figure S6 appears cut off, and as I result I believe it is missing a legend describing what background color and point color and shape means (it’s not in the caption).

PLOS authors have the option to publish the peer review history of their article (what does this mean?). If published, this will include your full peer review and any attached files.

Reviewer #1: No

Reviewer #2: No

---

## [Editor Report · Acceptance letter]

16 Jan 2025

Dear Dr. Chen,

We are delighted to inform you that your manuscript, "Order among chaos: high throughput MYCroplanters can distinguish interacting drivers of host infection in a highly stochastic system," has been formally accepted for publication in PLOS Pathogens.

Best regards,

Sumita Bhaduri-McIntosh

Editor-in-Chief

PLOS Pathogens

orcid.org/0000-0003-2946-9497

Michael Malim

Editor-in-Chief

PLOS Pathogens

orcid.org/0000-0002-7699-2064